# Retention Depolarization in Recommender System

## ABSTRACT

Repeated risk minimization is a popular choice in real-world recommender systems driving their recommendation algorithms to adapt to user preferences and trends. However, numerous studies have shown that it exacerbates retention disparities among user groups, resulting in polarization within the user population. Given the primary objective of improving long-term user engagement in most industrial recommender systems and the significant commercial benefits from a diverse user population, enforcing retention fairness across user population is therefore crucial. Nonetheless, this goal is highly challenging due to the unknown dynamics of user retention (e.g., when a user would abandon the system) and the simultaneous aim to maximize the experience of every user.

In this paper, we propose ReFair, the first computational framework that continuously improves recommendation algorithms while ensuring long-term retention fairness in the entire user population. ReFair alternates between environment learning (i.e., estimate the user retention dynamics) and fairness constrained policy improvement with respect to the estimated environment, while effectively handling uncertainties in the estimation. Our solution provides strong theoretical guarantees for long-term recommendation performance and retention fairness violation. Empirical experiments on two real-world recommendation datasets also demonstrate its effectiveness in realizing these two goals.

## KEYWORDS

recommender system, fairness, depolarization

### ACM Reference Format:

Anonymous Author(s). 2023. Retention Depolarization in Recommender System. In *Proceedings of ACM Conference (Conference'17)*. ACM, New York, NY, USA, 14 pages. https://doi.org/10.1145/nnnnnnn.nnnnnnn

## 1 INTRODUCTION

To continuously adapt to changing user preferences, evolving trends of content popularity, and dynamic market conditions, most industrial recommender systems actively update their algorithms by regularly incorporating new training data, e.g., user feedback on the recommendations. One widely-adopted approach is the repeated risk minimization (RRM) procedure [16], where the algorithm is updated by minimizing empirical loss on newly collected data, and then deployed to gather new data for next-step training. The procedure is repeated for iterative algorithmic refinement over time.

Previous studies [11, 32] show that RRM tends to exacerbate the performance disparities among different groups of users, leading to polarization within the user population. To make it more explicit, consider news recommendation to two user groups, one favoring international news and another favoring entertainment news, while the latter group is much larger in size. Due to the focus on minimizing overall loss in RRM, the recommendation algorithm tends to prioritize the optimization for the larger group's preference for entertainment news, resulting in a higher error rate and diminishing user experience in the smaller group that prefers international news. This, in turn, makes users in the smaller group more likely to abandon the system, which further shrinks its group size and their impact on the overall training objective of RRM. Consequently, the recommendation algorithm faces even greater challenges in capturing the preferences of the smaller group, i.e., a death spiral.

Thus, given the primary objective to improve long-term user engagement in industrial recommender systems and the significant commercial benefits from a diverse user population, it is crucial to depolarize and equalize retention across different user groups. This also forms a new notion of user-side fairness, which we refer to as *retention fairness*. However, previous research on user-side fairness in recommender system [14, 24, 27, 28, 30] fails to address this type of long-term fairness, primarily because of their focus on balancing instantaneous performance a user (or user groups) receives, measured by specific metrics, such as the prediction error disparity across different user groups at each time step of RRM. Yet, equalizing recommendation performance based on instantaneous metrics does not suggest equalized long-term retention [23, 40]. Instead, enforcing retention fairness necessitates a forward-looking approach that minimizes the disparity in retention across user groups over time, while the recommendation algorithm is being improved to maximize the recommendation utility. This goal is highly non-trivial, since the retention dynamics of different users are unknown and closely tied to the recommendations provided.

In this paper, we present a learning framework that iteratively updates its recommendation algorithm while enforcing long-term retention fairness, named as ReFair. To tackle the aforementioned challenges, we develop a model-based reinforcement learning (RL) solution, where we estimate the environment model to assist policy learning. At a high level, ReFair iteratively executes two steps: 1) estimate an individual user's reward feedback and retention after he/she takes a system-provided recommendation, and 2) improve the recommendation algorithm to maximize user satisfaction subject to retention fairness across user groups, both for the long-term. But because the estimations may be inaccurate, relying solely on the estimated environment can lead to sub-optimal recommendation performance [4, 8, 35] and also have no guarantee on the retention fairness. To address these issues, we propose a surrogate optimization approach that explicitly considers the uncertainty of the estimated environment. It incorporates an extra bonus to encourage exploration, while relaxing the retention fairness constraint based on the uncertainty of environment model estimation.

Policy gradient is then leveraged to solve the surrogate optimization problem. Theoretical analysis demonstrates that REFAIR achieves a sub-linear regret on cumulative reward and retention fairness violation, under a linear environment assumption. Experiments on two real-world datasets further demonstrate REFAIR's effectiveness in optimizing user satisfaction and ensuring retention fairness, both in a long term. All codes and data can be found in https://anonymous.4open.science/r/ReFair-BAB7.

In summary, our contributions are as follows:

- We introduce a model-based reinforcement learning solution that continuously improves recommendation quality while enforcing retention fairness in a long term.

- To address the inaccuracies in the estimated environment model, we propose a surrogate optimization approach that introduces additional exploration bonus and a soft constraint relaxation to counter uncertainty in environment model estimation.

- We demonstrate the effectiveness of our proposed framework through both theoretical analysis and empirical experiments on two real-world recommendation datasets.

## 2 PRELIMINARY

While the specific model architectures may vary, industrial recommendation systems commonly employ a repeated risk minimization (RRM) procedure to regularly incorporate new training data, e.g., user feedback on the latest recommendations, so as to provide up-to-date recommendations tailored to users' evolving preferences [16]. Formally, let $\pi_{\vartheta_t}$ denote the recommendation policy at time $t$ parameterized by $\vartheta_t$, and $\mathcal{H}_t$ denote the user interactions gathered by the previously deployed recommendation policy $\pi_{\vartheta_t}$, RRM updates the recommendation policy at time $t + 1$ to minimize the following loss:

$$\pi_{\vartheta_{t+1}} = \arg\min_{\vartheta} \mathbb{E}_{(u,a)\sim\mathcal{H}_t}[\ell(u, a; \vartheta_t)] \tag{1}$$

where $\ell(u, a; \vartheta_t)$ represents the loss associated with the interaction pair of user $u$ and recommendation $a$ from policy $\pi_{\vartheta_t}$. Various types of loss $\ell(u, a; \vartheta)$ have been explored in practice, such as cross-entropy loss [37], RL-based loss [5, 6], etc.

Numerous studies have shown that the RRM procedure in Eq. (1) exacerbates retention disparities among different user groups. As shown in previous studies [11, 32], minority groups, with limited observations in $\mathcal{H}_t$ and thus less impact on the training objective in Eq. (1), tend to experience worse recommendation quality when $\pi_{\vartheta_{t+1}}$ is deployed. Worse still, discouraged by subpar recommendations, such users are more likely to leave the platform, resulting in a further disparity in the sizes of training samples in $\mathcal{H}_{t+1}$ and their influence on the RRM training at time $t + 1$. This creates a detrimental feedback loop where minority groups suffer progressively inferior user experiences, leading to a continued shrinkage in group sizes.

While it is crucial to thoroughly investigate and address fairness concerns on long-term user retention, it has been overlooked in previous research on user-side fairness in recommender systems [14, 24, 27, 28, 30]. Previous research primarily focuses on enforcing fairness at a single step of RRM with respect to the instantaneous metric using collected data so far, such as minimizing prediction error gaps across user groups on $\mathcal{H}_t$. This however has no guarantee

on mitigating retention disparities in a long term, since smaller errors on an instantaneous metric (e.g., prediction error at time $t$) do not imply higher long-term user retention at all [23, 40]. This motivates our proposed framework for recommendation algorithm optimization subject to the retention fairness constraint.

## 3 REFAIR: OUR APPROACH WITH THEORETICAL GUARANTEES

In this section, we present REFAIR, a framework to iteratively improve recommendation algorithm/policy while enforcing retention fairness over time. As the first work of this type, we choose to focus on addressing retention fairness between two user groups in this paper, but the developed framework and algorithm can be easily extended to scenarios with multiple user groups.

As we focus on continuously improving the recommendation policy over time, we formulate the problem using a Markov Decision Process (MDP) and introduce an absorbing state $o$ to explicitly capture users' retention dynamics. Specifically, we construct the following MDP $\mathcal{M} = (\mathcal{S}, \mathcal{A}, \mathbb{P}, r, \rho_0, \gamma)$ where

- $\mathcal{S}$: a continuous state space describing the latent states of users. Here $s_{u,t} = o$ suggests user $u$ leaves the platform at time $t$. Otherwise, $s_{u,t}$ encodes the latent state at time $t$ of user $u$ based on the his/her historical interactions with the system (e.g., capturing and summarizing overall satisfaction of the system so far).

- $\mathcal{A}$: a discrete action space, containing all the recommendation candidates.

- $\mathbb{P} : \mathcal{S} \times \mathcal{A} \times \mathcal{S} \rightarrow \mathbb{R}$ is the state transition probability, where $\mathbb{P}(o|s_{u,t}, a)$ denotes the probability that user $u$ leaves the platform after being recommended content $a$ under state $s_{u,t}$.

- $r : \mathcal{S} \times \mathcal{A} \rightarrow \mathbb{R}$ is the reward function, where $r(s_{u,t}, a) \in [0, 1]$ represents the reward (e.g., rating) that the system obtains when recommending content $a$ to user $u$ at time $t$. Notably, once the user leaves the platform, no further reward can be obtained, i.e., $r(o, a) = 0$ for all $a \in \mathcal{A}$.

- $\rho_0$: the initial user state distribution at $t = 0$.

- $\gamma$: the discount factor for future rewards.

We employ two MDPs to represent two distinct user groups, such as those interested in entertainment news and international news in our previous example, respectively. These two MDPs share the same state space, action space, and discount factor, but they can differ in terms of initial state distributions, transitions, and reward functions. We use superscripts $g \in \{a, b\}$ to denote the two groups. For example, $\rho_0^a$ and $\rho_0^b$ denote the initial state distributions of the two groups, respectively. To simplify our notations, variables without superscripts are utilized to represent the entire user population. For example, $\rho_0$ denotes the initial state distribution over all users. At each time step $t = 0, 1, ...T$, the system samples an item from its latest recommendation policy $\pi_{\vartheta_t}$, which is a probability distribution over all recommendation candidates. The sampled item, denoted as $a_{u,t} \sim \pi_{\vartheta_t}(\cdot|s_{u,t})$, is then recommended to user $u$. Our goal is to find the $\pi_{\vartheta_t}$ that maximizes the cumulative reward from all users and concurrently realizes retention fairness between the two groups of users. This is formulated into the following optimization

problem,

$$\max_{\pi_\vartheta} \quad \mathbb{E}_{s_{u,t} \sim d_t, \pi_\vartheta} \left[ \sum_{k=0}^{T-t} \gamma^k \cdot r\left(s_{u,t+k}, a_{u,t+k}\right) \right] \tag{2}$$

$$\left| \mathbb{E}_{s_{u,t} \sim d_t^a, \pi_\vartheta} \left[ \mathbb{P}(\neq o | s_{u,t}, a_{u,t}) \right] - w \cdot \mathbb{E}_{s_{u,t} \sim d_t^b, \pi_\vartheta} \left[ \mathbb{P}(\neq o | s_{u,t}, a_{u,t}) \right] \right| \le \epsilon,$$

In the above, the constraint captures the *retention fairness* requirement we impose at each time step $t$:

- $d_t(\cdot)$, $d_t^a(\cdot)$ and $d_t^b(\cdot)$ denote the state distribution in the whole population, user group $a$ and $b$ at time $t$, respectively.

- The hyper-parameter $w$ is pre-defined to capture varying degrees of fairness requirements. For example, $w = 1$ indicates an equalization of retention between the two groups.

- $\mathbb{P}(\neq o | s_{u,t}, a_{u,t})$ is an abbreviation for $\mathbb{P}(s_{u,t+1} \neq o | s_{u,t}, a_{u,t})$, which represents the probability of user $u$ under state $s_{u,t}$ chooses to stay on the platform after being recommended to content $a_{u,t}$.

Under the context of recommender system optimization subject to retention fairness constraints, conventional value-based or policy-based reinforcement learning (RL) solutions become infeasible, as we cannot afford policy training by directly interacting with the environment (i.e., the users). More specifically, applying a policy that is not well optimized can cause deviations in the interaction trajectory from being satisfactory, leading to either poor cumulative rewards or significant violations of retention fairness. In particular, once a user abandons the platform, there is no way for the system to get the user back. In other words, our problem is not episodic: once initiated, we can never restart from the initial state.

Thus, we appeal to a model-based RL solution, which builds an environment model based on the estimated transition dynamics ($\hat{\mathbb{P}}_t$) and reward function ($\hat{r}_t$) at time $t$. The recommendation policy is then learned under the estimated environment. However, directly substituting the ground-truth reward function $r$ and transition dynamics $\mathbb{P}$ with their estimates to optimize Eq.(2) is not feasible. This is because the estimates cannot be perfect. Relying solely on the estimated rewards can lead to sub-optimal recommendation performance [4, 8, 35], while enforcing fairness based on the inaccurate estimations of transition dynamics can be misleading and thus fail to effectively ensure retention fairness.

As a result, we have to explicitly factor uncertainty of the estimated environment model into policy optimization, which closely couples the two iterative steps in REFAIR:

- **Environment learning.** Provide the estimated reward function $\hat{r}_t$ and transition dynamics $\hat{\mathbb{P}}_t$, along with the associated uncertainties required in the subsequent policy optimization step.

- **Policy optimization with the estimated environment.** Update the recommendation policy $\pi_{\vartheta_t}$ under the estimated environment ($\hat{r}_t, \hat{\mathbb{P}}_t$) at time $t$, with respect to the uncertainty of the estimated environment.

In the following, we start our discussion from policy learning, which imposes requirements for environment model learning.

## 3.1 Policy Improvement with An Estimated Environment

We devise a surrogate optimization problem that guides policy learning under the estimated environment, subject to the environment model's estimation uncertainty. The surrogate optimization problem is constructed based on Eq.(2) by introducing an exploration bonus term and relaxation of fairness constraint.

**Exploration Bonus.** Relying solely on the estimated reward $\hat{r}_t$ without considering its inaccuracy can mislead policy learning, e.g., overlook better recommendation policies under the ground-truth reward. A provably effective approach is to learn from calibrated rewards, i.e., the principle of optimism in the face of uncertainty [1, 12]. Specifically, we calibrate the estimated reward function with an exploration bonus term $b_t(\cdot, \cdot)$ that captures uncertainties in environment estimation and assigns higher values to currently under-explored actions as follows,

$$\tilde{r}_t(s, a) = \hat{r}_t(s, a) + b_t(s, a). \tag{3}$$

To make this calibration valid, $b_t(\cdot, \cdot)$ is required to consistently overestimate (i.e., be larger than) the true rewards [4, 8], which can be formally defined in the following:

**Definition 3.1** (Validity of exploration bonus $b_t$). A exploration bonus $b_t : \mathcal{S} \times \mathcal{A} \to \mathbb{R}$ is valid if, for $\forall s \in \mathcal{S}, a \in \mathcal{A}$, the following condition holds:

$$\left| \hat{r}_t(s, a) - r(s, a) + \gamma \left( \hat{\mathbb{P}}_t(\cdot | s, a) - \mathbb{P}(\cdot | s, a) \right) \hat{V}_t \right| \le b_t(s, a),$$

where $\hat{V}_t(s)$ denotes the value of state $s$ in the environment ($\tilde{r}_t, \hat{\mathbb{P}}_t$). And $\mathbb{P}(\cdot | s, a)\hat{V}_t = \sum_{s'} \mathbb{P}(s' | s, a)\hat{V}_t(s')$.

**Constraint Relaxation.** To address the influence of inaccuracies in the estimated transition dynamics $\hat{\mathbb{P}}_t$ when measuring retention fairness, we propose to relax the fairness constraint in Eq.(2):

$$\left| \mathbb{E}_{s_{u,t} \sim d_t^a, \pi_\vartheta} \left[ \hat{\mathbb{P}}_t(\neq o | s_{u,t}, a_{u,t}) \right] - w\mathbb{E}_{s_{u,t} \sim d_t^b, \pi_\vartheta} \left[ \hat{\mathbb{P}}_t(\neq o | s_{u,t}, a_{u,t}) \right] \right| \le c_t. \tag{4}$$

where $c_t$ represents the relaxation due to the uncertainty in the estimated $\hat{\mathbb{P}}_t$. As time progresses and more data is collected, the relaxation factor $c_t$ decreases along with the decreasing uncertainty in the estimated transition dynamics $\hat{\mathbb{P}}_t$. This allows the relaxed constraint to gradually approaches the desired ground-truth constraint in Eq.(2). Therefore the total violation of retention fairness during policy learning depends on the rate of shrinkage of $c_t$.

In addition to its rate of shrinkage, $c_t$ also needs to ensure that the set of policies satisfying the relaxed constraint include the optimal policy at time $t$, even if the estimated dynamics $\hat{\mathbb{P}}_t$ are not accurate yet. This requirement is referred to as the *compatibility* of $c_t$ [8]. Let $\pi_t^*$ represent the optimal policy for Eq.(2). We can formally define the compatibility of $c_t$ as follows,

**Definition 3.2** (Compatibility of $c_t$). $c_t$ is compatible, if $\pi_t^*$ is included in the policy set that satisfies the constraint, for all $t$:

$$\left| \mathbb{E}_{s_{u,t} \sim d_t^a, \pi_t^*} \left[ \hat{\mathbb{P}}_t(\neq o | s_{u,t}, a_{u,t}) \right] - w\mathbb{E}_{s_{u,t} \sim d_t^b, \pi_t^*} \left[ \hat{\mathbb{P}}_t(\neq o | s_{u,t}, a_{u,t}) \right] \right| \le c_t \tag{5}$$

**Surrogate optimization problem.** Let

$$\tilde{Q}_t(s_{u,t}, a) = \tilde{r}_t(s_{u,t}, a) + \mathbb{E}_{s_{u,t+1}} \left[ \sum_{k=1}^{T-t} \gamma^k \cdot \tilde{r}_t(s_{u,t+k}, a_{u,t+k}) \right]$$

denote the Q-function under the calibrated rewards in Eq.(3). Together with the uncertainty-driven relaxed constraint in Eq.(4), we

then learn $\pi_{\vartheta_t}$ by optimizing the following surrogate problem of Eq.(2) at time $t$:

$$\max_{\pi_\vartheta} \quad \mathbb{E}_{s_{u,t}\sim d_t,\pi_\vartheta}\left[\hat{Q}_t(s_{u,t}, a_{u,t})\right] \tag{6}$$

$$\left|\mathbb{E}_{s_{u,t}\sim d_t^a,\pi_\vartheta}\left[\hat{\mathbb{P}}_t(\neq o|s_{u,t}, a_{u,t})\right] - w\mathbb{E}_{s_{u,t}\sim d_t^b,\pi_\vartheta}\left[\hat{\mathbb{P}}_t(\neq o|s_{u,t}, a_{u,t})\right]\right| \le c_t.$$

Various constrained policy optimization algorithms [2, 34] can be employed to solve Eq.(6). And later we prove that solving this surrogate optimization problem leads to sublinear regret in recommendation performance and sublinear cumulative fairness constraint violation. In this work, we choose to customize a primal-dual gradient update procedure named FOCOPS [34], due to its stability and clear physical interpretations. The detailed implementation will be provided in Section 4.

**Extension to multiple user groups.** To extend ReFair to scenarios with $K$ user groups, we only need to formulate the retention fairness constraints among $K$ groups. A straightforward approach is to enforce retention fairness between every pair of groups as in Eq.(4), resulting in $K(K-1)/2$ constraints. The same constrained policy optimization algorithms as developed in this paper can then be directly applied to obtain the recommendation policy.

## 3.2 Environment Learning

In this section, we delve into the details of learning the environment model and deriving the valid exploration bonus term $b_t$ and compatible constraint relaxation $c_t$ accordingly.

To theoretically analyze the performance difference between learning through the surrogate optimization problem in Eq.(6) and the ideal optimization problem in Eq.(2), we assume the following linear structure in the ground-truth reward function $r$ and transition dynamics $\mathbb{P}$:

$$r(s_{u,t}, a_{u,t}) = \langle\theta_*, \phi(s_{u,t}, a_{u,t})\rangle$$
$$\mathbb{P}(s|s_{u,t}, a_{u,t}) = \langle\mu_*^s, \phi(s_{u,t}, a_{u,t})\rangle \tag{7}$$

where $\phi$ is a known state-action feature map $\phi : \mathcal{S} \times \mathcal{A} \to \mathbb{R}^d$; $\theta_* \in \mathbb{R}^d$ and $\mu_*^s \in \mathbb{R}^d$ are the unknown ground-truth parameters associated with the reward and transition dynamics for state $s$, respectively. Without loss of generality, we assume $\|\phi(s,a)\| \le 1$ for all $(s,a) \in \mathcal{S} \times \mathcal{A}$, $\|\theta_*\| \le \sqrt{d}$, and $\|v^\top\mu_*\| \le \sqrt{d}$ for any vector $v$ over $\mathcal{S}$ with $\|v\|_\infty \le 1$. Here, $\mu_*$ represents the stacked vector $\mu_*^s$ across $\mathcal{S}$. For simplicity, we will also use $r_{s,a}$ to represent $r(s,a)$ in the subsequent description.

At time $t$, the logged user-item interactions from previous timesteps are denoted as $D_t = \{\{s_{u,i}, a_{u,i}, s_{u,i+1}, r_{u,i}\}_{i=0}^{t-1}\}_u$. Based on the estimated model parameters $\hat{\theta}_t$ and $\hat{\mu}_t^s$, the reward and transition dynamics can be computed as follows:

$$\hat{r}_t(s_{u,t}, a_{u,t}) = \langle\hat{\theta}_t, \phi(s_{u,t}, a_{u,t})\rangle,$$
$$\hat{\mathbb{P}}_t(s|s_{u,t}, a_{u,t}) = \langle\hat{\mu}_t^s, \phi(s_{u,t}, a_{u,t})\rangle$$

where

$$\hat{\theta}_t = \arg\min_\theta \sum_u \sum_{i=0}^{t-1} \left(\theta^\top\phi(s_{u,i}, a_{u,i}) - r_{u,i}\right)^2 + \kappa\|\theta\|^2,$$

$$\hat{\mu}_t^s = \arg\min_\mu \sum_u \sum_{i=0}^{t-1} (\mu^\top\phi(s_{u,i}, a_{u,i}) - \mathbb{1}_{s_{u,i+1}=s})^2 + \kappa\|\mu\|^2.$$

The optimization problems above have closed-form solutions:

$$\hat{\theta}_t = \sum_u \sum_{i=0}^{t-1} r_{u,t}\phi(s_{u,i}, a_{u,i})^\top\Lambda_t^{-1}$$

$$\hat{\mu}_t^s = \sum_u \sum_{i=0}^{t-1} \mathbb{1}_{s_{u,i+1}=s}\phi(s_{u,i}, a_{u,i})^\top\Lambda_t^{-1} \tag{8}$$

with

$$\Lambda_t = \sum_{i=0}^{t-1}\sum_u \phi(s_{u,i}, a_{u,i})\phi(s_{u,i}, a_{u,i})^\top + \kappa I_{d\times d}.$$

Based on the closed-form solution in Eq.(8), we can derive the valid exploration bonus $b_t$ and compatible constraint relaxation $c_t$ in the following lemma.

**Lemma 3.3.** *Denote $\epsilon_{u,t}^s = \mathbb{P}(s|s_{u,t}, a_{u,t}) - \mathbb{1}_{s_{u,t+1}=s}$. Assume the linear structure in Eq.(7) holds, and $\mathbb{E}[\epsilon_{u,t}^s|D_t] = 0, \forall s$. For a fixed $\varsigma \in (0,1)$, with probability at least $1-\varsigma$, for all $t$, $s$ and $a$, the following constructions of exploration bonus $b_t$ and compatible $c_t$ are valid,*

$$b_t(s,a) = (\beta_1^t + \beta_2^t)\|\phi(s,a)\|_{\Lambda_t^{-1}}$$

$$c_t = \beta_3^t\mathbb{E}_{s_{u,t}\sim d_t^a,\pi_\vartheta}\left[\|\phi(s_{u,t}, a_{u,t})\|_{\Lambda_t^{-1}}\right]$$
$$+ w\beta_3^t\mathbb{E}_{s_{u,t}\sim d_t^b,\pi_\vartheta}\left[\|\phi(s_{u,t}, a_{u,t})\|_{\Lambda_t^{-1}}\right]$$

*where $\beta_1^t = \tilde{O}(d\sqrt{\log(Ut)})$, $\beta_2^t = O(\sqrt{d\log(tU)})$, $\beta_3^t = O(\sqrt{d\log(tU)})$, and $U$ denotes the number of users in the system.*

**Slow switching.** We adopt a slow switching technique to reduce computation overhead in updating the environment model [35]. The idea is that we only update the environment model when enough new data has been collected, via checking the determinant of the covariance matrix $\Lambda_t$. Specifically, assume the most recent model update happened at time $t$, we choose to update the model at time $t'$ only if $\det(\Lambda_{t'}) \ge 2\det(\Lambda_t)$.

**Theoretical Analysis.** To theoretically inspect the performance of learning through surrogate optimization, we consider the following regret:

- For recommendation performance, we track cumulative regret bound in $T$ rounds:

$$\text{Regret}(T) = \sum_{t=0}^{T-1}\left(\mathbb{E}_{s_{u,t}}[V^*(s_{u,t})] - \mathbb{E}_{s_{u,t}}\left[\sum_{k=0}^{T-t}\gamma^k r(s_{u,t+k}, a_{u,t+k})\right]\right)$$

where $\mathbb{E}_{s_{u,t}}[V^*(s_{u,t})]$ denotes the expected cumulative rewards obtained by following the optimal policy at time $t$.

- For retention fairness, we consider cumulative violation of fairness over T rounds:

$$\text{C}_{\text{reg}}(T) = \sum_{t=0}^{T-1}\left|\mathbb{E}_{s_{u,t}\sim d_t^a,\pi_{\vartheta_t}}\left[\mathbb{P}(\neq o|s_{u,t}, a_{u,t})\right]\right.$$
$$\left. -w\cdot\mathbb{E}_{s_{u,t}\sim d_t^b,\pi_{\vartheta_t}}\left[\mathbb{P}(\neq o|s_{u,t}, a_{u,t})\right]\right|$$

Based on the exploration bonus $b_t$ and compatible constraint relaxation $c_t$ derived in Lemma 3.3, along with the slow switching technique, we can bound the regret in long-term recommendation performance and total retention fairness violation of the recommendation policy learned through the surrogate optimization problem in Eq.(6) in the following theorem.

**Theorem 3.4.** *Assume the assumption in Lemma 3.3 holds. With probability at least $1 - \varsigma$, learning through the surrogate optimization problem in Eq.(6) has the following upper bounds on the cumulative regret for recommendation performance and total violation of retention fairness over $T$ rounds:*

$$Regret(T) \leq \tilde{O}\left(\sqrt{d^3 T \log(1/\varsigma)} \log(UT)\right)$$

$$C_{reg}(T) \leq O\left((1+w)d\sqrt{T \log(1/\varsigma)} \log(UT)\right)$$

## 4 PRACTICAL IMPLEMENTATION

In this section, we present our practical approach for solving the surrogate optimization problem defined in Eq.(6).

To ensure monotonic improvement, we adopt the local policy search method [19] to iteratively improve the policy. Additionally, we replace the second constraint in Eq.(6) with a square norm constraint, which is differentiable everywhere, facilitating more efficient and stable gradient-based optimization. As a result, we reformulate the surrogate optimization problem at time $t$ in Eq.(6) as follows:

$$\max_{\pi_\vartheta} \quad \mathbb{E}_{s_{u,t} \sim d_t, \pi_\vartheta}\left[\hat{Q}_t(s_{u,t}, a_{u,t})\right] \tag{9}$$

$$\left(\mathbb{E}_{s_{u,t} \sim d_t^a, \pi_\vartheta}\left[\hat{\mathbb{P}}_t(\neq o|s_{u,t}, a_{u,t})\right] - w\mathbb{E}_{s_{u,t} \sim d_t^b, \pi_\vartheta}\left[\hat{\mathbb{P}}_t(\neq o|s_{u,t}, a_{u,t})\right]\right)^2 \leq c_t^2$$

$$D_{\mathrm{KL}}(\pi_\vartheta \| \pi_{\vartheta_{t-1}}) \leq \sigma$$

We then employ a primal-dual gradient update algorithm FO-COPS [34] to solve Eq.(9), which first finds the optimal update policy in the nonparameterized policy space and then projects it back into the parametric policy space.

**Find the Optimal Policy Update.** Let

$$C_{t,\pi} = \mathbb{E}_{s_{u,t} \sim d_t^a, \pi}\left[\hat{\mathbb{P}}_t(\neq o|s_{u,t}, a_{u,t})\right] - w\mathbb{E}_{s_{u,t} \sim d_t^b, \pi}\left[\hat{\mathbb{P}}_t(\neq o|s_{u,t}, a_{u,t})\right]. \tag{10}$$

By following a similar procedure as FOCOPS [34], we can show that the optimal solution of Eq.(9) takes the following form:

- If user $u$ is in group $a$:

$$\pi^*(a|s_{u,t}) = \frac{\pi_{\vartheta_{t-1}}(a|s_{u,t})}{Z_{\lambda,v}(s_{u,t})} \exp\left(\frac{1}{v}\hat{Q}_t(s_{u,t}, a) - 2\frac{\lambda}{v}C_{t,\pi^*}\hat{\mathbb{P}}_t(\neq o|s_{u,t}, a)\right)$$

- If user $u$ is in group $b$:

$$\pi^*(a|s_{u,t}) = \frac{\pi_{\vartheta_{t-1}}(a|s_{u,t})}{Z_{\lambda,v}(s_{u,t})} \exp\left(\frac{1}{v}\hat{Q}_t(s_{u,t}, a) + 2w\frac{\lambda}{v}C_{t,\pi^*}\hat{\mathbb{P}}_t(\neq o|s_{u,t}, a)\right)$$

where $Z_{\lambda,v}(s_{u,t})$ ensures $\pi^*(\cdot|s_{u,t})$ is a valid probability distribution, i.e., $\sum_{a \in \mathcal{A}} \pi^*(\cdot|s_{u,t}) = 1$. $\lambda$ and $v$ are the Lagrangian multiplier of the first and second constraint in Eq. (9) respectively, and they correspond to the solutions of the dual problem. See Appendix 8.1 for a complete derivation.

The optimal solution carries an intuitive physical interpretation. For example, when group $a$ is the current disadvantage group (i.e., the fairness violation $C_{t,\pi_\vartheta} \leq 0$), the Q-value of actions with high retention probability is boosted. This interpretation holds in a similar manner when group $b$ becomes the disadvantage group.

**Calculation of Policy Gradient.** To obtain the policy gradients for optimizing Eq.(9), we project $\pi^*$ into the parameterized policy

---

**Algorithm 1** ReFair.

Initialize $\Lambda_0 = \kappa I$, $Z = \det(\Lambda_0)$.
**for** $t = 0, 1, \ldots, T$ **do**
  Calculate $\Lambda_t$.
  **if** $\det(\Lambda_t) \geq 2Z$ **then**
    Estimate the environment, i.e., $\tilde{r}_t$, $\hat{\mathbb{P}}_t$, $b_t$ and $c_t$.
    $Z = \det(\Lambda_t)$.
  **end if**
  Update $\lambda$ using Eq.(12).
  **for** $K$ epochs **do**
    Update recommendation policy using Eq.(11).
  **end for**
  Take action with respect to $\pi_{\vartheta_t}$ and log interactions.
**end for**

---

space by minimizing the following loss:

$$\mathcal{L}(\vartheta) = \mathbb{E}_{s_{u,t} \sim d_t}\left[D_{\mathrm{KL}}(\pi_\vartheta \| \pi^*)[s_{u,t}]\right].$$

This results in the following policy gradients:

$$\nabla_\vartheta \mathcal{L}(\vartheta) = \mathbb{E}_{s_{u,t} \sim d_t}\left[\nabla_\vartheta D_{\mathrm{KL}}(\pi_\vartheta \| \pi^*)[s_{u,t}]\right],$$

where

$$\nabla_\vartheta D_{\mathrm{KL}}(\pi_\vartheta \| \pi^*)[s_{u,t}] \approx \nabla_\vartheta D_{\mathrm{KL}}(\pi_\vartheta \| \pi_{\vartheta_{t-1}})[s_{u,t}] \tag{11}$$

$$- \frac{1}{v}\mathbb{E}_{a \sim \pi_{\vartheta_{t-1}}}\left[\frac{\nabla_\vartheta \pi_\vartheta(a|s_{u,t})}{\pi_{\vartheta_{t-1}}(a|s_{u,t})}\left(\hat{Q}_t(s_{u,t}, a) + 2w^g\lambda C_{t,\pi_\vartheta}\hat{\mathbb{P}}_t(\neq o|s_{u,t}, a)\right)\right].$$

Here $w^g = -1$ if the user $u$ is in group $a$ and $w^g = w$ if the user is in group $b$. Details of derivation can be found in Lemma 8.1 in the appendix.

**Dynamically adjust constraint strength.** $\lambda$ and $v$ control the strength of the two constraints in Eq.(9). However, directly solving the dual problem to obtain specific values for $\lambda$ and $v$ is computationally impractical for large state/action spaces as it requires to calculate $Z_{\lambda,v}(s_{u,t})$.

Following previous work [34], we treat the parameter $v$ as a fixed hyperparameter during training, since it plays a role similar to the temperature term utilized in maximum entropy reinforcement learning [39]. Furthermore, as the strong duality holds, we can optimize the dual problem by applying gradient descent with respect to $\lambda$ to determine the current optimal fairness constraint strength. This leads to the following update rule for $\lambda$:

$$\lambda \leftarrow \mathrm{proj}_\lambda[\lambda - \alpha(c_t^2 - C_{t,\pi_\vartheta}^2)] \tag{12}$$

The projection operator $\mathrm{proj}_\lambda$ projects $\lambda$ back to $[0, \lambda^{\max}]$; and $\alpha$ is the step size. The detailed derivation of the policy gradients with respect to $\lambda$ can be found in Appendix 8.1.

The update of the fairness constraint strength, as shown in Eq.(12), also takes into account the uncertainty in environment estimation. The strength is only increased when the constraint violation exceeds the current estimation uncertainty. The implementation of ReFair is summarized in Algorithm 1.

## 5 EXPERIMENTS

In this section, we empirically evaluate the performance of Re-Fair on two real-world recommendation datasets. We assess the effectiveness of the algorithm in terms of both long-term recommendation quality and retention fairness among groups.

## 5.1 Experiment Setup

**Datasets.** We adopt the following two benchmark recommendation datasets with long-term user activity records.

- **ML-1M dataset.**[1] The dataset consists of the activity records of 6,040 users spanning from year 2000 to 2003, encompassing approximately 1 million ratings for around 3,900 movies from the online movie recommendation service MovieLens. We set $r(u, a) = 1$, if user $u$ gives movie $a$ a rating greater than 3, otherwise $r(u, a) = 0$. To ensure data quality, we only retain users who have provided more than 10 positive rewards and movies that have received more than 50 ratings.

- **30Music dataset [21].** The dataset consists of listening and playlist data from 45K users, including 31,351,954 play events on 5.6 million tracks, within a 1-year time window starting from January, 2014 on Last.FM. We take $r(u, a) = 1$ if user $u$ completes listening to a song $a$ at least once, and $r(u, a) = 0$ otherwise. To ensure data quality, we adopt the 100-core setting [33], i.e., discarding users and tracks with less than 100 interactions.

The statistics of the two datasets are summarized in Table 1.

**Simulated Environment.** To evaluate the long-term performance of a recommendation algorithm, it is crucial to allow the algorithm to interact with users. Following previous work [7], we focus on evaluating the algorithms on the two benchmark datasets by training an environment simulator to mimic an interactive environment.

The architecture of the environment simulator is depicted in Figure 3 in the appendix. Following previous work [6, 7], we employ a Recurrent Neural Network (RNN) to capture the temporal dynamics of a user's state transition. At each time step $t$, the user's state $s_{u,t}$ is constructed by concatenating two components: (1) embedded user features and ID information, and (2) the RNN's output, which summarizes the user's interaction history from previous $t - 1$ steps. The RNN recursively feeds its output at time $t - 1$, the recommended item to user $u$ at time $t$, and the corresponding user feedback as input. The element-wise product between $s_{u,t}$ and the item embedding of item $a$ yields $\phi(s_{u,t}, a)$. This vector is then linearly projected to obtain the reward $r(s_{u,t}, a)$ and the probability of user $u$ abandoning the platform after receiving the recommendation $a$, i.e., $\mathbb{P}(o|s_{u,t}, a)$.

The environment simulator is trained by minimizing its error in predicting the reward and the probability of platform abandonment recorded in the dataset. Considering the sparsity of the abandonment signal among users, we define the event of a user's departure from the platform as after receiving a recommendation there is no further interaction within two weeks in the ML-1M dataset or 12 hours in the 30Music dataset, when constructing the training dataset. If a user returns to the platform after the specified time period (two weeks for ML-1M or 12 hours for 30Music), they are treated as a new user with the same user features but an empty interaction history.

During the evaluation process, after the algorithm recommends item $a$ to the user under state $s_{u,t}$, the representation $\phi(s_{u,t}, a)$ and reward $r(s_{u,t}, a)$ generated by the simulator are presented to the algorithm. Then we sample $s_{u,t+1} \sim \mathbb{P}(\cdot|s_{u,t}, a)$; if $s_{u,t+1} = o$, the

---

[1] https://grouplens.org/datasets/movielens/1m/

**Table 1: Statistics of Datasets.**

| Dataset | #Users | #Items | #Interactions |
|---------|--------|--------|---------------|
| ML-1M | 6,040 | 3,883 | 1,000,209 |
| 30Music | 45,000 | 5,675,143 | 31,351,954 |

user is considered as leaving the system afterwards; otherwise the recommendation process continues.

In our experiments, we cluster users into two distinct preference groups based on user feature/ID embeddings learned from the environment simulator. We focus on the setting where $w = 1$, defined in Eq.(2), which corresponds to equalizing the long-term user retention of the two user groups.

**Baselines.** We compare ReFair with the following baselines.

- **RRM**: the repeated risk minimization procedure, as discussed in Section 2, involves iteratively minimizing the loss on one-step intermediate feedback to obtain the updated recommendation policy. This approach is commonly utilized in real-world recommendation systems to enable continuous algorithm update.

- **DRO [11]**: a repeated risk minimization procedure, based on distributional robust optimization. It minimizes the maximum loss among groups at each time step. This approach has been used to mitigate retention disparity in [11].

- **RRM-Fair [27, 28]**: a repeated risk minimization procedure that minimizes the difference of training loss among different groups at each time step to realize fairness.

- **RL-UnFair**: a model-based RL algorithm that maximizes the estimated Q-value for policy learning without incorporating exploration bonuses or considering retention-level fairness at each time step [40].

- **RL-DM**: the model-based RL algorithm that learns a policy $\pi$ by directly maximizing the objective function:

$$\mathbb{E}_\pi[\hat{Q}_t(s_{u,t}, a_{u,t}) - \lambda C_{t,\pi}]$$

where $\lambda$ is a fixed hyper-parameter that balances the trade-off between fairness and recommendation performance (in analogy to the Lagrange multiplier method). The term $C_{t,\pi}$ is defined in Eq.(10) and captures the difference in user retention between two groups based on the estimated dynamics at time $t$ (i.e., $\hat{\mathbb{P}}_t$).

Both **RL-UnFair** and **RL-DM** can be viewed as heuristic solutions leveraging our RL formulation.

**Evaluation Metrics.** We evaluate all algorithms in terms of their long-term recommendation performance and retention fairness.

For recommendation performance, we consider two metrics: (1) *Cumulative reward*, computed by $\mathbb{E}_{s_{u,0} \sim \rho} \left[ \sum_{t=0}^{T} \gamma^t r(s_{u,t}, a_{u,t}) \right]$; and (2) *Active rate@T*, which measures the ratio of active users at time $T$ to the total number of users. Higher values for both metrics indicate better recommendation performance in the long run.

For retention fairness, we utilize the metric *Retention disparity*, which quantifies the ratio of the retention probability between the advantage group (with higher retention probability) and the disadvantage group (with lower retention probability). A value closer to 1 indicates a more fair algorithm in terms of long-term retention. We adopt ratio as a measure of disparity, following [7].

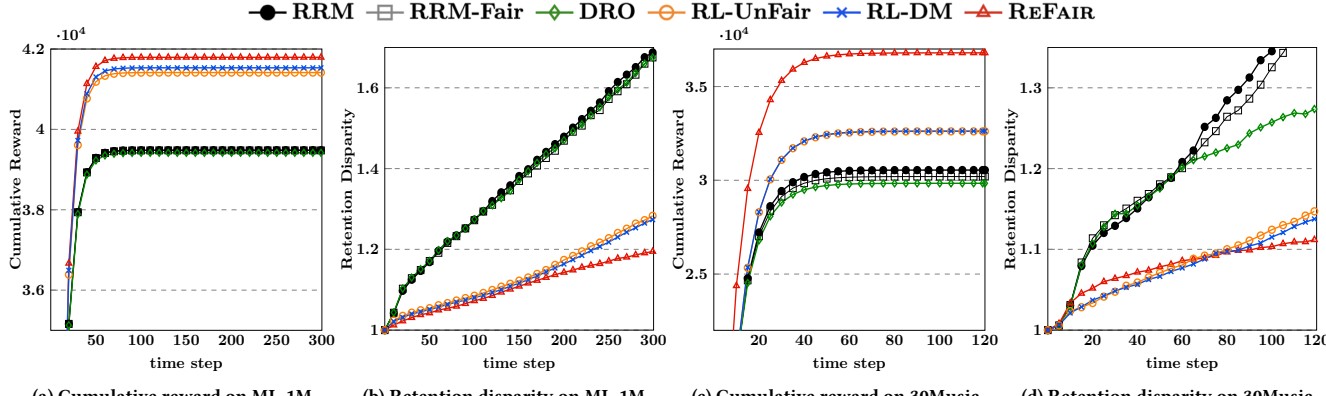

(a) Cumulative reward on ML-1M    (b) Retention disparity on ML-1M    (c) Cumulative reward on 30Music    (d) Retention disparity on 30Music

Figure 1: Experiment results regarding cumulative reward and retention disparity on the two real-world datasets.

Table 2: Experiment results regarding Active rate@T.

| Algorithm | ML-1M | 30Music |
|-----------|-------|---------|
| RRM | 0.3870 | 0.2017 |
| RRM-Fair | 0.3869 | 0.1982 |
| DRO | 0.3873 | 0.3096 |
| RL-UnFair | 0.7103 | 0.5900 |
| RL-DM | 0.7151 | 0.5943 |
| ReFair | **0.7718** | **0.6247** |

## 5.2 Experiment Results

We executed each algorithm for 10 runs with different random seeds. The average cumulative reward and retention disparity of each algorithm on the two datasets during the whole recommendation process are presented in Figure 1, while the Active rate@T performance is reported in Table 2.

• *Long-term dynamics matters.* Compared to algorithms that also use our RL formulation, the line of RRM-based algorithms (i.e., RRM, DRO, RRM-Fair), focusing only on instantaneous objectives at single time step, exhibited much lower cumulative rewards (Figure 1a and 1c), lower active rates thus higher abandonment probabilities (Table 2), as well as higher retention disparities significantly increasing over time (Figure 1b and 1d). This highlights the necessity and importance of considering long-term dynamics of user satisfaction and retention in recommendation policy learning.

• *Short-term fairness intervention still polarizes.* As shown by the performance of DRO and RRM-Fair in Figure 1b and 1d, simply enforcing fairness with respect to the instantaneous metric still leads to high and increasing retention disparities over time, i.e., polarized user population in the long run.

• *Enforcing retention fairness under uncertainty depolarizes and improves recommendation performance.* The approaches of RL-UnFair (without explicitly imposing a retention fairness constraint) or RL-DM (enforcing fairness without considering uncertainty in transition dynamics estimation) fail in managing retention disparity over time, as demonstrated in Figure 1b and 1d. In contrast, ReFair explicitly factors the environment model estimation uncertainty into its policy learning and hence achieves better retention fairness, as evidenced by the lowest and mostly converged retention disparity. Moreover, ReFair achieves the highest cumulative reward and active rate@T, indicating stronger long-term recommendation performance.

Table 3: Effects of exploration bonus and fairness constraint strength adjustment regarding Active rate@T.

| Algorithm | RL-UnFair | ReFair-OnlyQ | ReFair-alpha0 | ReFair |
|-----------|-----------|--------------|---------------|--------|
| ML-1M | 0.7103 | 0.7085 | 0.7412 | **0.7718** |
| 30Music | 0.5900 | 0.6212 | 0.6232 | **0.6247** |

## 5.3 Ablation Studies

To gain a comprehensive understanding of ReFair, we conducted ablation studies to examine the effects of two crucial factors in its design: (1) the impact of the exploration bonus, and (2) the effect of dynamically adjusting the strength of the fairness constraint by updating $\lambda$ through dual optimization. We introduce two variants of ReFair respectively:

- **ReFair-onlyQ**: ReFair that maximizes Q-values based on exploration bonus-calibrated reward without considering retention fairness.

- **ReFair-alpha0**: ReFair without dynamically adjusting the strength of the fairness constraint, i.e., $\alpha = 0$ in Eq (12).

• *Exploration bonus enhances recommendation performance.* From Figure 2a and 2b, and Table 3, it can be observed that enhancing reward via exploration bonus (ReFair-onlyQ) results in higher cumulative reward and active rate@T compared to RL-UnFair, which directly works under the estimated environment model. This emphasizes the importance of exploration in compensating the inaccuracies in environment estimation. However, without explicitly enforcing retention fairness, ReFair-onlyQ experiences higher and increasing retention disparity over time compared to ReFair, as shown in Figure 2c and 2d.

• *Dynamically adjusting the strength of fairness constraint provides better fairness control.* Compared to ReFair-alpha0, ReFair demonstrates improved control over retention disparity by dynamically adjusting the strength of the retention fairness constraint through the update of $\lambda$, leading to better retention fairness.

## 6 RELATED WORK

Our work introduces the first framework that enables continuous improvement of recommendation algorithms while simultaneously maintaining long-term retention fairness. The following two lines of work are most related to this work.

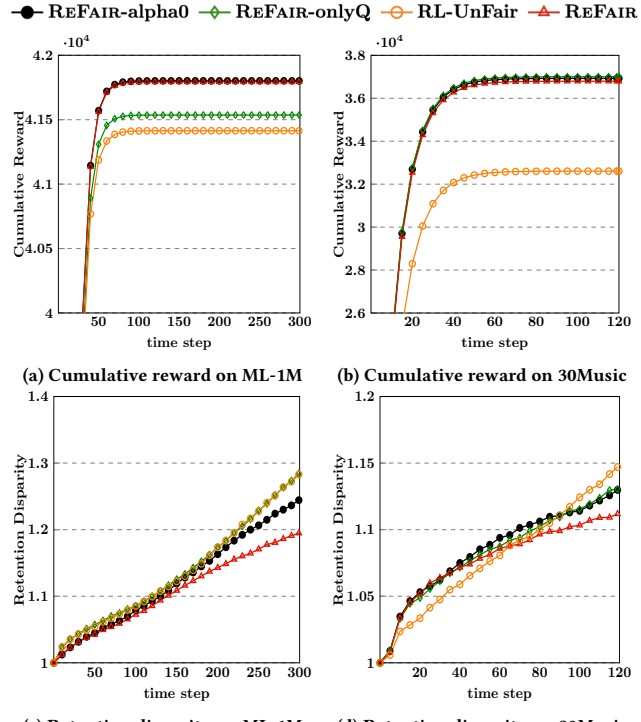

(a) Cumulative reward on ML-1M

(b) Cumulative reward on 30Music

(c) Retention disparity on ML-1M

(d) Retention disparity on 30Music

**Figure 2: Effects of exploration bonus and fairness constraint strength adjustment on reward and retention disparity.**

**Fairness in Recommender System.** Fairness, being a fundamental concept in trustworthy machine learning, has garnered significant attention in the field of recommender systems [18, 22]. Extensive research has been conducted to explore and address fairness concerns in recommendation scenarios, which can be broadly categorized into three types based on the stakeholders involved: 1) User-side fairness, which aims to balance the quality of recommendation among different users (or user groups); 2) Item-side fairness [17, 38], which focuses on equalizing exposure of different item groups in the recommendation list; and, 3) Multiple-sided fairness [25], which considers fairness concerns from multiple perspectives, such as user-side, item-side, and sometimes even producer-side fairness. User-side fairness can be further classified into individual fairness and group fairness. Individual fairness [15, 26] adopts a counterfactual notion and aims to ensure similar users receive recommendations of similar quality. On the other hand, user-side group fairness focuses on balancing recommendation performance among different user groups [7, 14, 24, 27, 28, 30]. Our work specifically addresses user-side group fairness. However, most existing work on user-side group fairness [14, 24, 27, 28, 30] primarily focuses on balancing group performance on instantaneous metrics (e.g., prediction error) at a specific time step of RRM. Unfortunately, these approaches cannot address retention fairness, which requires equalized user retention over time [23, 32, 40]. Our experiment results in Section 5.2 further support this argument.

Under the context of long-term fairness in recommender systems, previous work [9, 10, 20] has primarily focused on leveraging reinforcement learning techniques to ensure long-term item-side fairness, i.e., equalizing exposure among different item groups in a long run. A recent study by Chi et al. [7] utilizes reinforcement learning to achieve equalized cumulative rewards across user groups. However, this approach requires the policy to directly interact with users for data collection and policy optimization. If the policy is not well optimized, especially in the early stage, it can lead to poor recommendations and seriously hurts user engagement with the platform. Once a user abandons the platform, there is no way for the system to regain their participation. In contrast, our work estimates an environment model for policy learning, which mitigates the risk of losing users' trust due to the poor recommendation quality.

**Long-term Fairness.** Some recent work [8, 29, 31] has approached long-term group fairness as a reinforcement learning problem, primarily focusing on a sequential binary decision making setting. However, these settings differ fundamentally from recommendation scenarios in two key aspects. Firstly, most of them [8, 29] focus on an episodic setting, which assumes the decision making process will repeatedly restart from the initial state, which is equivalent to having all users back to the recommender system when a predefined time horizon comes to its end. But in reality, it is not possible to regain users who have left the platform. Secondly, in this line of research, the actions themselves (such as granting a loan) directly determine the level of fairness (such as the difference in loan approval rates among user groups). In contrast, a recommendation decision, i.e., what to recommend, involves unknown impacts on in user retention among groups. This significantly complicates policy learning and constraint satisfaction.

## 7 CONCLUSION

In this paper, we introduce a novel fairness notion concerning long-term retention across different user groups, driven by the necessity of diverse long-term user engagement in recommender systems. We propose the first framework that enables continuous improvement of recommendation algorithms while enforcing retention fairness. To tackle the challenge of unknown user retention dynamics, we propose REFAIR, a model-based reinforcement learning approach that alternates between environment estimation and fairness control with respect to the uncertainty of environment estimation. Our rigorous theoretical analysis demonstrates that REFAIR achieves a sub-linear regret on cumulative reward and constraint violation, when the underlying MDP possesses a linear structure. Furthermore, empirical experiments conducted on two real-world recommendation datasets validate the effectiveness of REFAIR in optimizing long-term user satisfaction and ensuring retention fairness.

In our theoretical analysis, we primarily focused on linear environments. As our future work, we plan to explore the extension of REFAIR to more complex environment assumptions, including kernel-based [35] and neural network-based [36] models. Additionally, in this work, the recommendation policy is learnt completely from the estimated environment. Given the effectiveness of offline RL algorithms [13], previously logged interactions can also be leveraged to jump start policy learning and further improve empirical performance. Furthermore, analyzing the tightness of our constraint relaxation and exploring new methods to further tighten it are also important for achieving further improvements on retention fairness.

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

## 7.1 Theoretical Proof.

In this section, we provide a detailed proof of the theorems and lemmas in Section 3.2. For simplicity, we use $\pi_t$ to represent $\pi_{\vartheta_t}$ in the following proof.

**Proof of Lemma 3.3:**

PROOF. We first show the validity of the exploration bonus.

***Validity of*** $b_t$. Recall that $\epsilon_{u,t}^s = \mathbb{P}(s|s_{u,t}, a_{u,t}) - \mathbb{1}_{s_{u,t+1}=s}$, and $\mathbb{E}[\epsilon_{u,t}^s|D_t] = 0, \forall s$. Under the linear structure assumption in Eq.(7), we have the following lemmas from previous work [1, 3].

**Lemma 7.1** (Uniform Convergence Results (Lemma 8.7 in [3]))**.** *Fix* $\varsigma \in (0,1)$, *for all* $t$, *all* $s, a$, *with probability at least* $1 - \varsigma$, *we have:*

$$\left|\left(\hat{\mathbb{P}}_t(\cdot|s,a) - \mathbb{P}(\cdot|s,a)\right)\hat{V}_t\right| \leq \beta_1^t \|\phi(s,a)\|_{\Lambda_t^{-1}},$$

*with* $\beta_1^t = \tilde{O}(\sqrt{d\log(tU)})$.

**Lemma 7.2** (Theorem 2 in [1]). *For a fixed $\varsigma \in (0,1)$, with probability at least $1 - \varsigma$, for all $t$, and all $s, a$, we have:*

$$|\hat{r}_t(s,a) - r(s,a)| \leq \|\phi(s,a)\|_{\Lambda_t^{-1}} \underbrace{\left(\sqrt{d\log(\frac{1+tU/\kappa}{\sigma})} + \kappa^{1/2}L\right)}_{\beta_2^t}$$

According the definition of valid exploration bonus in Definition 3.1, we have:

$$\left|\hat{r}_t(s,a) - r(s,a) + \gamma\left(\hat{\mathbb{P}}_t(\cdot|s,a) - \mathbb{P}(\cdot|s,a)\right)\hat{V}_t\right|$$
$$\leq |\hat{r}_t(s,a) - r(s,a)| + \gamma\left|\left(\hat{\mathbb{P}}_t(\cdot|s,a) - \mathbb{P}(\cdot|s,a)\right)\hat{V}_t\right|$$
$$\leq \beta_2^t\|\phi(s,a)\|_{\Lambda_t^{-1}} + \gamma\beta_1^t\|\phi(s,a)\|_{\Lambda_t^{-1}}.$$

This conclude the validity of the exploration bonus $b_t$.

**Compatible $c_t$.** Next we show the compatibility of the derived $c_t$. According to the Definition 3.2, we have

$$\left|\mathbb{E}_{s_{u,t}\sim d_t^a,\pi_t^*}\left[\hat{\mathbb{P}}_t(\neq o|s_{u,t},a_{u,t})\right] - w\mathbb{E}_{s_{u,t}\sim d_t^b,\pi_t^*}\left[\hat{\mathbb{P}}_t(\neq o|s_{u,t},a_{u,t})\right]\right|$$

$$\leq \left|\mathbb{E}_{s_{u,t}\sim d_t^a,\pi_t^*}\left[\sum_{s'\neq o}\hat{\mathbb{P}}_t(s'|s_{u,t},a_{u,t}) - \mathbb{P}(s'|s_{u,t},a_{u,t})\right]\right|$$

$$+ w\left|\mathbb{E}_{s_{u,t}\sim d_t^b,\pi_t^*}\left[\sum_{s'\neq o}\hat{\mathbb{P}}_t(s'|s_{u,t},a_{u,t}) - \mathbb{P}(s'|s_{u,t},a_{u,t})\right]\right|$$

$$+ \underbrace{\left|\mathbb{E}_{s_{u,t}\sim d_t^a,\pi_t^*}\left[\mathbb{P}(\neq o|s_{u,t},a_{u,t})\right] - w\mathbb{E}_{s_{u,t}\sim d_t^b,\pi_t^*}\left[\mathbb{P}(\neq o|s_{u,t},a_{u,t})\right]\right|}_{\leq \epsilon, \text{ for } \pi_t^*}$$

$$\tag{13}$$

Let $\tilde{\beta}_3^t = \left(\kappa\sqrt{d} + 3\sqrt{d\log\left(\frac{1+tU/\kappa}{\sigma}\right)}\right)$. According to Lemma 7.3, we can bound the first two terms as follows:

$$\left|\mathbb{E}_{s_{u,t}\sim d_t^a,\pi_t^*}\left[\sum_{s'\neq o}\hat{\mathbb{P}}_t(s'|s_{u,t},a_{u,t}) - \mathbb{P}(s'|s_{u,t},a_{u,t})\right]\right|$$

$$\leq \mathbb{E}_{s_{u,t}\sim d_t^a,\pi_t^*}\left[\left|\sum_{s'\neq o}\hat{\mathbb{P}}_t(s'|s_{u,t},a_{u,t}) - \mathbb{P}(s'|s_{u,t},a_{u,t})\right|\right]$$

$$\leq \tilde{\beta}_3^t\mathbb{E}_{s_{u,t}\sim d_t^a,\pi_t^*}\left[\|\phi(s_{u,t},a_{u,t})\|_{\Lambda_t^{-1}}\right]$$

Let $\varrho_{\max} = \max_{s_{u,t},a}\frac{\pi_t^*(a|s_{u,t})}{\pi_t(a|s_{u,t})}$ and $\beta_3^t = \varrho_{\max}\cdot\tilde{\beta}_3^t$, with importance sampling, Eq.(13) can be rewritten as:

$$\left|\mathbb{E}_{s_{u,t}\sim d_t^a,\pi_t^*}\left[\hat{\mathbb{P}}_t(\neq o|s_{u,t},a_{u,t})\right] - w\mathbb{E}_{s_{u,t}\sim d_t^b,\pi_t^*}\left[\hat{\mathbb{P}}_t(\neq o|s_{u,t},a_{u,t})\right]\right|$$

$$\leq \underbrace{\beta_3^t\mathbb{E}_{s_{u,t}\sim d_t^a,\pi_t}\left[\|\phi(s_{u,t},a_{u,t})\|_{\Lambda_t^{-1}}\right] + w\beta_3^t\mathbb{E}_{s_{u,t}\sim d_t^b,\pi_t}\left[\|\phi(s_{u,t},a_{u,t})\|_{\Lambda_t^{-1}}\right]}_{c_t}$$

This concludes the whole proof. $\qquad\qquad\square$

**Lemma 7.3.** *For a fixed $\sigma \in (0,1)$, with probability at least $1 - \sigma$, for all $t$, and all $s, a$, we have:*

$$\left|\sum_{s'\neq o}\hat{\mathbb{P}}_t(s'|s,a)] - \mathbb{P}(s'|s,a)\right| \leq \left(\kappa\sqrt{d} + 3\sqrt{d\log\left(\frac{1+tU/\kappa}{\sigma}\right)}\right)\|\phi(s,a)\|_{\Lambda_t^{-1}}$$

PROOF. Define $V_o : \mathcal{S} \to [0,1]$ with $V_o(o) = 0$, otherwise $V_o(s) = 1, \forall s \neq o$. Let $\hat{\mu}_t$ and $\epsilon_{u,i}$ represent the stacked vectors of $\hat{\mu}_t^s$ and $\epsilon_{u,i}^s = \mathbb{P}(s|s_{u,i},a_{u,i}) - \mathbb{1}_{s_{u,i+1}=s}$ across $\mathcal{S}$, respectively. Then we have:

$$\left|\sum_{s'\neq o}\hat{\mathbb{P}}_t(s'|s,a)] - \mathbb{P}(s'|s,a)\right| = \left|\left(\hat{\mathbb{P}}_t(\cdot|s,a) - \mathbb{P}(\cdot|s,a)\right)V_o\right|$$

$$= |(\hat{\mu}_t\phi(s,a) - \mu_*\phi(s,a))\cdot V_o|$$

$$\overset{(1)}{\leq} \left|\kappa\phi(s,a)^T\Lambda_t^{-1}(\mu_*)^TV_o\right| + \left|\sum_{i=0}^{t-1}\sum_u\phi(s,a)^T\Lambda_t^{-1}\phi(s_{u,i},a_{u,i})(\epsilon_{u,i})^TV_o\right|$$

$$\leq \kappa\sqrt{d}\|\phi(s,a)\|_{\Lambda_t^{-1}} + \|\phi(s,a)\|_{\Lambda_t^{-1}}\cdot\|\sum_{i=0}^{t-1}\sum_u\phi(s_{u,i},a_{u,i})(\epsilon_{u,i})^TV_o\|_{\Lambda_t^{-1}}$$

where inequality (1) is according to the Lemma 8.3 in [3]. Moreover, Lemma 8.4 in [3] shows that:

$$\|\sum_{i=0}^{t-1}\sum_u\phi(s_{u,i},a_{u,i})(\epsilon_{u,i})^TV_o\|_{\Lambda_t^{-1}} \leq 3\sqrt{\ln\frac{\det(\Lambda_t)^{1/2}\det(\kappa I)^{-1/2}}{\sigma}}$$

$$\leq 3\sqrt{d\log\left(\frac{1+tU/\lambda}{\kappa}\right)}.$$

Thus we concludes the proof. $\qquad\qquad\square$

**Proof of Theorem 3.4:**

PROOF. We first prove the cumulative regret bound over $T$ rounds. We denote $\hat{V}_t^\pi(s_{u,t})$ as the value function of $\pi$ at state $s_{u,t}$ with estimated reward function $\tilde{r}_t$ in Eq.(3) and estimated transition dynamics $\hat{\mathbb{P}}_t$. Let $V_t(s_{u,t})$ represent the expected cumulative reward, starting from time $t$ and following the derived policy sequence $\pi_{\vartheta_t}, \pi_{\vartheta_{t+1}}, ...$, on the ground-truth MDP with the true reward function $r$ and transition dynamics $\mathbb{P}$. And we define $\Delta_t = \mathbb{E}_{s_{u,t}\sim d_t}[V^*(s_{u,t})] - \mathbb{E}_{s_{u,t}\sim d_t}[V_t(s_{u,t})]$.

Then we have:

$$\text{Regret}(T) = \sum_{t=0}^{T-1}\left(\mathbb{E}_{s_{u,t}}[V^*(s_{u,t})] - \mathbb{E}_{s_{u,t}}\left[\sum_{k=0}^{T-t}\gamma^kr(s_{u,t+k},a_{u,t+k})\right]\right)$$

$$= \sum_{t=0}^{T-1}\Delta_t. \tag{14}$$

We then get:

$$\Delta_t \overset{(a)}{\leq} \mathbb{E}_{s_{u,t}\sim d_t}[\hat{V}_t^*(s_{u,t})] - \mathbb{E}_{s_{u,t}\sim d_t}[V_t(s_{u,t})]$$

$$\overset{(b)}{\leq} \mathbb{E}_{s_{u,t}\sim d_t}[\hat{V}_t^{\pi_t}(s_{u,t})] - \mathbb{E}_{s_{u,t}\sim d_t}[V_t(s_{u,t})] \tag{15}$$

where the inequality (a) is due to the optimism proved in Lemma 7.5, and inequality (b) arises from the fact that $\pi_t$ is the optimal solution for Eq.(6).

Recall that:

$$\hat{Q}_t^{\pi_t}(s_{u,t},a_{u,t}) = \tilde{r}_t(s_{u,t},a_{u,t}) + \gamma\hat{\mathbb{P}}_t(\cdot|s_{u,t},a_{u,t})\hat{V}_t^{\pi_t},$$

thus we have:

$$\hat{Q}_t^{\pi_t}(s_{u,t}, a_{u,t}) - Q_t(s_{u,t}, a_{u,t})$$

$$= \tilde{r}_t(s_{u,t}, a_{u,t}) + \gamma \hat{\mathbb{P}}_t(\cdot|s_{u,t}, a_{u,t})\hat{V}_t^{\pi_t} - r(s_{u,t}, a_{u,t}) - \gamma \mathbb{P}_t(\cdot|s_{u,t}, a_{u,t})V_{t+1}$$

$$= \hat{r}_t(s_{u,t}, a_{u,t}) - r(s_{u,t}, a_{u,t}) + \gamma(\hat{\mathbb{P}}_t(\cdot|s_{u,t}, a_{u,t}) - \mathbb{P}(\cdot|s_{u,t}, a_{u,t}))\hat{V}_t^{\pi_t}$$

$$+ \gamma \mathbb{P}(\cdot|s_{u,t}, a_{u,t})\left(\hat{V}_t^{\pi_t} - V_{t+1}\right) + b_t(s_{u,t}, a_{u,t}).$$

Let

$$\text{BELL}_{u,t} = \hat{r}_t(s_{u,t}, a_{u,t}) - r(s_{u,t}, a_{u,t}) + \gamma(\hat{\mathbb{P}}_t(\cdot|s_{u,t}, a_{u,t}) - \mathbb{P}(\cdot|s_{u,t}, a_{u,t}))\hat{V}_t^{\pi_t},$$

and take expectation regarding policy $\pi_t$ to select $a_{u,t}$ and state, we have:

$$\mathbb{E}_{s_{u,t} \sim d_t}[\hat{V}_t^{\pi_t}(s_{u,t})] - \mathbb{E}_{s_{u,t} \sim d_t}[V_t(s_{u,t})]$$

$$\leq \mathbb{E}_{s_{u,t} \sim d^t, \pi_t}\left[\text{BELL}_{u,t}\right] + \mathbb{E}_{s_{u,t} \sim d^t, \pi_t}\left[b_t(s_{u,t}, a_t)\right]$$

$$+ \gamma \mathbb{E}_{s_{u,t+1} \sim d_{t+1}}[\hat{V}_t^{\pi_t}(s_{u,t+1}) - V_{t+1}(s_{u,t+1})]$$

$$\leq \mathbb{E}_{s_{u,t} \sim d^t, \pi_t}\left[\text{BELL}_{u,t}\right] + \mathbb{E}_{s_{u,t} \sim d^t, \pi_t}\left[b_t(s_{u,t}, a_t)\right]$$

$$+ \gamma \mathbb{E}_{s_{u,t+1} \sim d_{t+1}}[\hat{V}_t^{\pi_t}(s_{u,t+1}) - \hat{V}_{t+1}^{\pi_{t+1}}(s_{u,t+1})]$$

$$+ \gamma \mathbb{E}_{s_{u,t+1} \sim d_{t+1}}[\hat{V}_{t+1}^{\pi_{t+1}}(s_{u,t+1}) - V_{t+1}(s_{u,t+1})].$$

Summarizing over $t = 0, \ldots, T-1$, we have:

$$\sum_{t=0}^{T-1} \mathbb{E}_{s_{u,t} \sim d_t}[\hat{V}_t^{\pi_t}(s_{u,t}) - V_t(s_{u,t})] \leq \sum_{t=0}^{T-1} \mathbb{E}_{s_{u,t} \sim d^t, \pi_t}\left[\text{BELL}_{u,t} + b_t(s_{u,t}, a_{u,t})\right]$$

$$+ \sum_{t=0}^{T-1} \gamma \mathbb{E}_{s_{u,t+1} \sim d_{t+1}}[\hat{V}_t^{\pi_t}(s_{u,t+1}) - \hat{V}_{t+1}^{\pi_{t+1}}(s_{u,t+1})]$$

$$+ \sum_{t=0}^{T-1} \gamma \mathbb{E}_{s_{u,t+1} \sim d_{t+1}}[\hat{V}_{t+1}^{\pi_{t+1}}(s_{u,t+1}) - V_{t+1}(s_{u,t+1})] \qquad (16)$$

For the last term, we have:

$$\sum_{t=0}^{T-1} \mathbb{E}_{s_{u,t+1} \sim d_{t+1}}[\hat{V}_{t+1}^{\pi_{t+1}}(s_{u,t+1}) - V_{t+1}(s_{u,t+1})]$$

$$\leq \sum_{t=0}^{T-1} \mathbb{E}_{s_{u,t} \sim d_t}[\hat{V}_t^{\pi_t}(s_{u,t}) - V_t(s_{u,t})] - (\hat{V}_0^{\pi_0}(s_{u,0}) - V_0(s_{u,0}))$$

$$+ (\hat{V}_T^{\pi_T}(s_{u,T}) - V_T(s_{u,T}))$$

$$\leq 2V_{\max} + \sum_{t=0}^{T-1} \mathbb{E}_{s_{u,t} \sim d_t}[\hat{V}_t^{\pi_t}(s_{u,t}) - V_t(s_{u,t})] \qquad (17)$$

Plugging Eq.(17) into Eq.(16), we have:

$$\sum_{t=0}^{T-1} \mathbb{E}_{s_{u,t} \sim d_t}[\hat{V}_t^{\pi_t}(s_{u,t}) - V_t(s_{u,t})] \leq \frac{1}{1-\gamma}\mathbb{E}[\sum_{t=0}^{T} \text{BELL}_{u,t} + b_t(s_{u,t}, a_{u,t})]$$

$$+ \frac{\gamma}{1-\gamma}\sum_{t=0}^{T-1} \mathbb{E}_{s_{u,t+1} \sim d_{t+1}}[\hat{V}_t^{\pi_t}(s_{u,t+1}) - \hat{V}_{t+1}^{\pi_{t+1}}(s_{u,t+1})] + \frac{2V_{\max}}{1-\gamma} \qquad (18)$$

For the second part, since there is in total $M(T)$ switches, we known that there are at most $M(T)$ non-zero terms, thus we have:

$$\sum_{t=0}^{T-1} \mathbb{E}_{s_{u,t+1} \sim d_{t+1}}[\hat{V}_t^{\pi_t}(s_{u,t+1}) - \hat{V}_{t+1}^{\pi_{t+1}}(s_{u,t+1})] \leq V_{\max}M(T).$$

Plugging in the upper bound of $M(T)$ in Lemma 7.4, we get:

$$\sum_{t=0}^{T-1} \mathbb{E}_{s_{u,t+1} \sim d_{t+1}}[\hat{V}_t^{\pi_t}(s_{u,t+1}) - \hat{V}_{t+1}^{\pi_{t+1}}(s_{u,t+1})] \leq V_{\max}M(T)$$

$$\leq \frac{1}{\log 2}V_{\max}d\log\left(\frac{d + UT/\kappa}{d}\right) + 1.$$

For the first term, we have:

$$\mathbb{E}[\sum_{t=0}^{T} \text{BELL}_t + b_t(s_{u,t}, a_{u,t})] \leq \mathbb{E}[\sum_{t=0}^{T} |\text{BELL}_t| + b_t(s_{u,t}, a_{u,t})]$$

$$\leq 2\mathbb{E}[\sum_{t=0}^{T} b_t(s_{u,t,a_t})] \leq 2\mathbb{E}\left[\sum_{t=0}^{T}(\beta_1^t + \beta_2^t)\|\phi(s_{u,t}, a_{u,t})\|_{\Lambda_t^{-1}}\right]$$

$$= 2(\beta_1^T + \beta_2^T)\mathbb{E}\left[\sum_{t=0}^{T}\|\phi(s_{u,t}, a_{u,t})\|_{\Lambda_t^{-1}}\right]$$

$$\leq 2(\beta_1^T + \beta_2^T)\sqrt{T\sum_{t=0}^{T}\|\phi(s_{u,t}, a_{u,t})\|_{\Lambda_t^{-1}}^2}$$

$$\leq 2(\beta_1^T + \beta_2^T)\sqrt{Td\log\left(\frac{d + TU/\kappa}{d}\right)}.$$

Recalling that $\beta_1^T = \tilde{O}(d\sqrt{\log(UT)})$ and $\beta_2^T = O(\sqrt{d\log(TU)})$, we can summarize the results as follows:

$$\text{Regret}(T) \leq \tilde{O}\left(\sqrt{d^3 T \log(1/\varsigma)}\log(UT)\right)$$

**Constrained Violation Bound.** Next we bound the average violation of constraints over T rounds.

$$C_{\text{reg}}(T)$$

$$= \sum_{t=0}^{T-1}\left|\mathbb{E}_{s_{u,t} \sim d_t^a, \pi_t}\left[\mathbb{P}(\neq o|s_{u,t}, a_{u,t})\right] - w\mathbb{E}_{s_{u,t} \sim d_t^b, \pi_t}\left[\mathbb{P}(\neq o|s_{u,t}, a_{u,t})\right]\right|$$

$$\leq \sum_{t=0}^{T-1}\left|\mathbb{E}_{s_{u,t} \sim d_t^a, \pi_t}\left[\hat{\mathbb{P}}_t(\neq o|s_{u,t}, a_{u,t})\right] - w\mathbb{E}_{s_{u,t} \sim d_t^b, \pi_t}\left[\hat{\mathbb{P}}_t(\neq o|s_{u,t}, a_{u,t})\right]\right|$$

$$+ \sum_{t=0}^{T-1}\mathbb{E}_{s_{u,t} \sim \bar{d}_t^a, \pi_t}\left[\left\|\sum_{s' \neq o}\mathbb{P}(s'|s_{u,t}, a_{u,t}) - \hat{\mathbb{P}}(s'|s_{u,t}, a_{u,t})\right\|\right]$$

$$+ w\sum_{t=0}^{T-1}\mathbb{E}_{s_{u,t} \sim \bar{d}_t^b, \pi_t}\left[\left\|\sum_{s' \neq o}\mathbb{P}(s'|s_{u,t}, a_{u,t}) - \hat{\mathbb{P}}(s'|s_{u,t}, a_{u,t})\right\|\right]$$

$$\leq 2c_t$$

And for $g \in \{a, b\}$, we have:

$$\mathbb{E}_{s_{u,t} \sim \bar{d}_t^g}[\sum_{t=0}^{T-1}\|\phi(s_{u,t}, a_{u,t})\|_{\Lambda_t^{-1}}] \leq \sqrt{Td\log\left(\frac{d + TU/\kappa}{d}\right)}$$

Recall that $\beta_3^T = O(\sqrt{d\log(TU)})$, thus the constraint violation over T rounds is bounded by:

$$C_{\text{reg}}(T) \leq O\left((1+w)d\sqrt{T\log(1/\varsigma)}\log(UT)\right)$$

Thus we conclude the proof. □

**Lemma 7.4.** *(Bounding the switches.) The total number of switches can be bounded as follows:*

$$M(T) \leq \frac{1}{\log 2} d \log\left(\frac{d + UT/\kappa}{d}\right) + 1$$

PROOF. On one hand, we have

$$\frac{\det(\Lambda_T)}{\det(\Lambda_0)} \geq \prod_{s=1}^{M(T)-1} \frac{\det(\Lambda_{t_{s+1}})}{\det(\Lambda_{t_s})} \geq 2^{M(T)-1}.$$

On the otherhand, we also have:

$$\frac{\det(\Lambda_T)}{\det(\Lambda_0)} = \det(\Lambda_0^{-1}\Lambda_T) \leq \left(\frac{\text{Tr}(\Lambda_0^{-1}\Lambda_T)}{d}\right)^d \leq \left(\frac{d + UT/\kappa}{d}\right)^d.$$

Combing above inequalities, we can get:

$$M(T) \leq \frac{1}{\log 2} d \log\left(\frac{d + UT/\kappa}{d}\right) + 1$$

□

**Lemma 7.5** (Optimism). *For any policy $\pi$, we have*

$$V^\pi(s_{u,t}) \leq \hat{V}_t^\pi(s_{u,t}), \qquad \forall u, t$$

.

PROOF. Recall that $\hat{V}_t$ is calculated with estimated reward function $\tilde{r}_t$ in Eq.(3) and estimated transition dynamics $\hat{\mathbb{P}}_t$. Since we focus on the finite horizon settings, thus $V^\pi(s_{u,T+1}) = \hat{V}_t^\pi(s_{u,T+1}) = 0$.

We prove this lemma through induction. Assuming the inductive hypothesis $V^\pi(s_{u,t+1}) \leq \hat{V}_t^\pi(s_{u,t+1})$, we have:

$$\hat{V}_t^\pi(s_{u,t}) = \mathbb{E}_\pi\left[\hat{r}_t(s_{u,t}, a_{u,t}) + b_{u,t} + \gamma \sum_{s_{u,t+1}} \hat{\mathbb{P}}_t(s_{u,t+1}|s_{u,t}, a_{u,t})\hat{V}_t^\pi(s_{u,t+1})\right]$$

$$\overset{(1)}{\geq} \mathbb{E}_\pi\left[r(s_{u,t}, a_{u,t}) + \gamma \sum_{s_{u,t+1}} \mathbb{P}(s_{u,t+1}|s_{u,t}, a_{u,t})\hat{V}_t^\pi(s_{u,t+1})\right]$$

$$\overset{(2)}{\geq} \mathbb{E}_\pi\left[r(s_{u,t}, a_{u,t}) + \gamma \sum_{s_{u,t+1}} \mathbb{P}(s_{u,t+1}|s_{u,t}, a_{u,t})V^\pi(s_{u,t+1})\right]$$

$$= V^\pi(s_{u,t})$$

Inequality (1) is due to the validity of $b_{u,t}$ in Definition 3.1, and inequality (2) is due to the inductive hypothesis $V^\pi(s_{u,t+1}) \leq \hat{V}_t^\pi(s_{u,t+1})$

□

# 8 APPENDIX

## 8.1 Detailed derivation of policy gradients of Eq.(9).

We first provide a detailed derivation for the optimal policy $\pi^*$ in Eq. (9). Let $\lambda$ and $v$ denote the Lagrangian multiplier of the first and second constraint in Eq. (9) respectively. Then we have

$$\mathcal{L}(\pi_\vartheta, \lambda, v) = \lambda c_t^2 + v\sigma + \mathbb{E}_{s_{u,t}, \pi_\vartheta}[\hat{Q}_t(s_{u,t}, a_{u,t})] - \lambda C_{t,\pi_\vartheta}^2 - vD_{\text{KL}}(\pi_\vartheta \| \pi_{\vartheta_{t-1}})$$

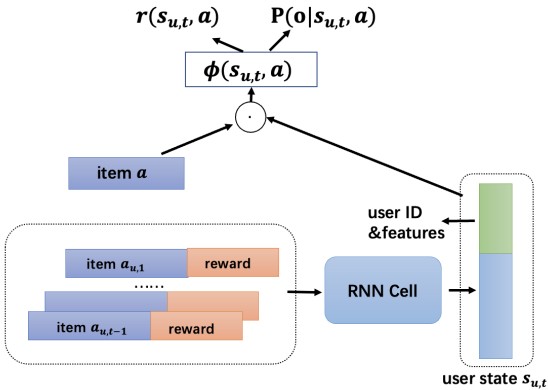

**Figure 3: Architecture of the environment simulator.**

We can observe $\mathcal{L}(\pi_\vartheta, \lambda, v)$ is linear with respect to $\pi_\vartheta$. Therefore, it follows that Slater's constraint qualification is satisfied and strong duality holds.

$$p^* = \max_{\pi_\vartheta} \min_{\lambda, v \geq 0} \mathcal{L}(\pi_\vartheta, \lambda, v) = \min_{\lambda, v \geq 0} \max_{\pi_\vartheta} \mathcal{L}(\pi_\vartheta, \lambda, v). \quad (19)$$

For the inner optimization problem, i.e.,

$$\max_{\pi_\vartheta} \mathcal{L}(\pi_\vartheta, \lambda, v)$$

$$s.t., \sum_a \pi_\vartheta(a|s_{u,t}) = 1, \pi_\vartheta(a|s_{u,t}) \geq 0$$

Similarly, denoting $\zeta$ as the Lagrange multiplier for the constraints $\sum_a \pi_\vartheta(a|s_{u,t}) = 1$, we have:

$$G(\pi_\vartheta) = \lambda c_t^2 - \lambda C_{t,\pi_\vartheta}^2 - \zeta\left(\sum_a \pi_\vartheta(a|s_{u,t}) - 1\right)$$

$$+ \mathbb{E}_{s_{u,t}}\left[\sum_a \pi_\vartheta(a|s_{u,t})\left(\hat{Q}_t(s_{u,t}, a) - v(\log \pi_\vartheta(a|s_{u,t}) - \log \pi_{\vartheta_{t-1}}(a|s_{u,t}))\right)\right]$$

Taking the derivative of $G(\pi)$ with respect to $\pi_\vartheta$ and setting it to zero, we can derive the following optimal policy:

- If user $u$ is in group $a$:

$$\pi^*(a|s_{u,t}) = \frac{\pi_{\vartheta_{t-1}}(a|s_{u,t})}{Z_{\lambda,v}(s_{u,t})} \exp\left(\frac{1}{v}\hat{Q}_t(s_{u,t}, a) - 2\frac{\lambda}{v}C_{t,\pi^*}\hat{\mathbb{P}}_t(\neq o|s_{u,t}, a)\right)$$

- If user $u$ is in group $b$:

$$\pi^*(a|s_{u,t}) = \frac{\pi_{\vartheta_{t-1}}(a|s_{u,t})}{Z_{\lambda,v}(s_{u,t})} \exp\left(\frac{1}{v}\hat{Q}_t(s_{u,t}, a) + 2w\frac{\lambda}{v}C_{t,\pi^*}\hat{\mathbb{P}}_t(\neq o|s_{u,t}, a)\right)$$

Let $w^g = -1$ if the user is in group $a$ and $w^g = w$ if the user is in group $b$. By substituting $\pi^*$ into Eq. (19), we obtain:

$$p^* = \min_{\lambda, v \geq 0} \lambda c_t^2 + v\sigma - \lambda C_{t,\pi^*}^2$$

$$+ \mathbb{E}_{s_{u,t}, \pi^*}\left[\hat{Q}_t(s_{u,t}, a) - v(\log \pi^*(a|s_{u,t}) - \log \pi_{\vartheta_{t-1}}(a|s_{u,t}))\right]$$

$$= \lambda c_t^2 + v\sigma - \lambda C_{t,\pi^*}^2$$

$$+ \mathbb{E}_{s_{u,t}, \pi^*}\left[v \log Z_{\lambda,v}(s_{u,t}) - 2\lambda w^g C_{t,\pi^*}\hat{\mathbb{P}}_t(\neq o|s_{u,t}, a)\right] \quad (20)$$

**Lemma 8.1.** *The gradients of*

$$\mathcal{L}(\vartheta) = \mathbb{E}_{s_{u,t} \sim d_t}[D_{\text{KL}}(\pi_\vartheta \| \pi^*)[s_{u,t}]]$$

*takes the form:*

$$\nabla_\vartheta \mathcal{L}(\vartheta) = \mathbb{E}_{s_{u,t} \sim d_t}[\nabla_\vartheta D_{\text{KL}}(\pi_\vartheta \| \pi^*)[s_{u,t}]],$$

with

$$\nabla_\vartheta D_{\mathrm{KL}}(\pi_\vartheta \| \pi^*)[s_{u,t}] = \nabla_\vartheta D_{\mathrm{KL}}(\pi_\vartheta \| \pi_{\vartheta_{t-1}})[s_{u,t}]$$

$$- \frac{1}{v} \mathbb{E}_{a \sim \pi_{\vartheta_{t-1}}} \left[ \frac{\nabla_\vartheta \pi_\vartheta(a|s_{u,t})}{\pi_{\vartheta_{t-1}}(a|s_{u,t})} \left( Q(s_{u,t}, a) + 2w^g \lambda C_{t,\pi_\vartheta} \hat{\mathbb{P}}_t(\neq o|s_{u,t}, a) \right) \right],$$

where $w^a = -1$ and $w^b = w$.

PROOF. We first have that :

$$D_{\mathrm{KL}}(\pi_\vartheta \| \pi^*)[s_{u,t}] = \underbrace{- \sum_a \pi_\vartheta(a|s_{u,t}) \log \pi^*(a|s_{u,t})}_{(A_1)}$$

$$+ \sum_a \pi_\vartheta(a|s_{u,t}) \log \pi_\vartheta(a|s_{u,t})$$

Let

$$w^g = \begin{cases} -1, & g = a \\ w. & g = b \end{cases}$$

For $A_1$, we have:

$$A_1 = - \sum_a \pi_\vartheta(a|s_{u,t}) \log \left( \frac{\pi_{\vartheta_{t-1}}(a|s_{u,t})}{Z_{\lambda,v}(s_{u,t})} \cdot \right.$$

$$\left. \exp \left( \frac{1}{v} Q(s_{u,t}, a) + 2w^g \frac{\lambda}{v} C_{t,\pi^*} \hat{\mathbb{P}}_t(\neq o|s_{u,t}, a) \right) \right)$$

$$\approx - \sum_a \pi_\vartheta(a|s_{u,t}) \log \pi_{\vartheta_{t-1}}(a|s_{u,t}) + \log Z_{\lambda,v}(s_{u,t})$$

$$- \frac{1}{v} \sum_a \pi_\vartheta(a|s_{u,t}) \left( Q(s_{u,t}, a) + 2w^g \lambda C_{t,\pi_\vartheta} \hat{\mathbb{P}}_t(\neq o|s_{u,t}, a) \right)$$

We then subtract the entropy term to recover the KL divergence:

$$D_{\mathrm{KL}}(\pi_\vartheta \| \pi^*)[s_{u,t}] = D_{\mathrm{KL}}(\pi_\vartheta \| \pi_{\vartheta_{t-1}})[s_{u,t}] + \log Z_{\lambda,v}(s_{u,t})$$

$$- \frac{1}{v} \mathbb{E}_{a \sim \pi_{\vartheta_{t-1}}} \left[ \frac{\nabla_\vartheta \pi_\vartheta(a|s_{u,t})}{\pi_{\vartheta_{t-1}}(a|s_{u,t})} \left( Q(s_{u,t}, a) + 2w^g \lambda C_{t,\pi_\vartheta} \hat{\mathbb{P}}_t(\neq o|s_{u,t}, a) \right) \right]$$

where the last equality we applied importance sampling to rewrite the expectation w.r.t $\pi_{\vartheta_{t-1}}$. Finally, taking the gradient on both sides, we will finish the proof. $\square$

**Update $\lambda$ and $v$.** Recall the dual optimization problem is derived as follows (Eq.(20)):

$$p^* = \min_{\lambda, v \geq 0} \lambda c_t^2 + v\sigma - \lambda C_{t,\pi^*}^2$$

$$+ \mathbb{E}_{s_{u,t}, \pi^*} \left[ v \log Z_{\lambda,v}(s_{u,t}) - 2\lambda w^g C_{t,\pi^*} \hat{\mathbb{P}}_t(\neq o|s_{u,t}, a) \right]$$

For simplicity, we denote

$$\tilde{Q}_t^g(s_{u,t}, a)) = \frac{1}{v} \hat{Q}_t(s_{u,t}, a) + 2w^g \frac{\lambda}{v} C_{t,\pi^*} \hat{\mathbb{P}}_t(\neq o|s_{u,t}, a).$$

We first focus on the derivative of $\pi^*$ regarding to the $\lambda$.

$$\frac{\partial \pi^*(a|s_{u,t})}{\partial \lambda}$$

$$= \frac{\pi_{\vartheta_{t-1}}(a|s_{u,t})}{Z_{\lambda,v}^2(s_{u,t})} \left[ Z_{\lambda,v}(s_{u,t}) \frac{\partial}{\partial \lambda} \exp(\tilde{Q}_t^g(s_{u,t}, a)) \right.$$

$$\left. - \exp(\tilde{Q}_t^g(s_{u,t}, a)) \frac{\partial Z_{\lambda,v}(s_{u,t})}{\partial \lambda} \right]$$

$$= \frac{\pi_{\vartheta_{t-1}}(a|s_{u,t})}{Z_{\lambda,v}(s_{u,t})} \exp(\tilde{Q}_t^g(s_{u,t}, a)) \frac{\partial \tilde{Q}_t^g(s_{u,t}, a)}{\lambda}$$

$$- \frac{\pi_{\vartheta_{t-1}}(a|s_{u,t})}{Z_{\lambda,v}(s_{u,t})} \exp(\tilde{Q}_t^g(s_{u,t}, a)) \frac{\partial \log Z_{\lambda,v}(s_{u,t})}{\lambda}$$

$$= 2 \frac{w^g}{v} C_{t,\pi^*} \hat{\mathbb{P}}_t(\neq o|s_{u,t}, a) \pi^*(a|s_{u,t}) - \pi^*(a|s_{u,t}) \frac{\partial \log Z_{\lambda,v}(s_{u,t})}{\lambda}$$

Thus the derivative of the last expectation term in $p^*$ with respect to $\lambda$ can be expressed as:

$$\frac{\partial}{\partial \lambda} \mathbb{E}_{s_{u,t}, \pi^*} \left[ v \log Z_{\lambda,v}(s_{u,t}) - 2\lambda w^g C_{t,\pi^*} \hat{\mathbb{P}}_t(\neq o|s_{u,t}, a) \right]$$

$$= \mathbb{E}_{s_{u,t}, \pi_{\vartheta_{t-1}}} \left[ \frac{\partial}{\partial \lambda} \frac{\pi^*(a|s_{u,t})}{\pi_{\vartheta_{t-1}}(a|s_{u,t})} \left( -2\lambda w^g C_{t,\pi^*} \hat{\mathbb{P}}_t(\neq o|s_{u,t}, a) \right. \right.$$

$$\left. \left. + v \log Z_{\lambda,v}(s_{u,t}) \right) \right]$$

$$= \mathbb{E}_{s_{u,t}, \pi_{\vartheta_{t-1}}} \left[ \frac{1}{\pi_{\vartheta_{t-1}}(a|s_{u,t})} \left( v\pi^*(a|s_{u,t}) \frac{\log Z_{\lambda,v}(s_{u,t})}{\partial \lambda} \right. \right.$$

$$- 2w^g \pi^*(a|s_{u,t}) C_{t,\pi^*} \hat{\mathbb{P}}_t(\neq o|s_{u,t}, a)$$

$$\left. \left. + \frac{\partial \pi^*(a|s_{u,t})}{\partial \lambda} \left( -2\lambda w^g C_{t,\pi^*} \hat{\mathbb{P}}_t(\neq o|s_{u,t}, a) + v \log Z_{\lambda,v}(s_{u,t}) \right) \right) \right].$$

$$= \mathbb{E}_{s_{u,t}, \pi^*} [2 \frac{w^g}{v} C_{t,\pi^*} \hat{\mathbb{P}}_t(\neq o|s_{u,t}, a) \left( v \log Z_{\lambda,v}(s_{u,t}) \right.$$

$$\left. - 2w^g \lambda C_{t,\pi^*} \hat{\mathbb{P}}_t(\neq o|s_{u,t}, a) \right)$$

$$- \frac{\partial \log Z_{\lambda,v}(s_{u,t})}{\partial \lambda} \left( v \log Z_{\lambda,v}(s_{u,t}) - 2w^g \lambda C_{t,\pi^*} \hat{\mathbb{P}}_t(\neq o|s, a) \right)$$

$$+ v \frac{\log Z_{\lambda,v}(s_{u,t})}{\partial \lambda} - 2w^g C_{t,\pi^*} \hat{\mathbb{P}}_t(\neq o|s_{u,t}, a)] \quad (21)$$

Also

$$\frac{\partial Z_{\lambda,v}(s_{u,t})}{\partial \lambda} = \frac{\partial}{\partial \lambda} \sum_a \pi_{\vartheta_{t-1}}(a|s_{u,t}) \exp(\tilde{Q}_t^g(s_{u,t}, a))$$

$$= \sum_a \pi_{\vartheta_{t-1}}(a|s_{u,t}) \exp(\tilde{Q}_t^g(s_{u,t}, a)) * \left( 2 \frac{w^g}{v} C_{t,\pi^*} \hat{\mathbb{P}}_t(\neq o|s_{u,t}, a) \right)$$

$$= \frac{Z_{\lambda,v}(s_{u,t})}{v} \mathbb{E}_{\pi^*} [2w^g C_{t,\pi^*} \hat{\mathbb{P}}_t(\neq o|s_{u,t}, a)].$$

Therefore,

$$\frac{\partial \log Z_{\lambda,v}(s_{u,t})}{\partial \lambda} = \frac{\partial Z_{\lambda,v}(s_{u,t})}{\partial \lambda} * \frac{1}{Z_{\lambda,v}(s_{u,t})}$$

$$= \frac{1}{v} \mathbb{E}_{\pi^*} [2w^g C_{t,\pi^*} \hat{\mathbb{P}}_t(\neq o|s_{u,t}, a)] \quad (22)$$

Thus,

$$\frac{\partial}{\partial \lambda} \mathbb{E}_{s_{u,t}, \pi^*} \left[ v \log Z_{\lambda,v}(s_{u,t}) - 2\lambda w^g C_{t,\pi^*} \hat{\mathbb{P}}_t(\neq o|s_{u,t}, a) \right] = 0 \quad (23)$$

The derivative of $\mathcal{L}(\pi^*, \lambda, v)$ w.r.t $\lambda$ is :

$$\frac{\partial \mathcal{L}(\pi^*, \lambda, v)}{\lambda} = c_t^2 - C_{t,\pi^*}^2 \tag{24}$$

With the optimized policy at time $t$ being $\pi_{\vartheta_t}$, the update rule for $\lambda$ is as follows:

$$\lambda \leftarrow \text{proj}_\lambda(\lambda - \alpha(c_t^2 - C_{t,\pi_{\vartheta_t}}^2)) \tag{25}$$

The projection operator $\text{proj}_\lambda$ projects $\lambda$ back to $[0, \lambda^{\max}]$. And $\alpha$ is the step size.

