# OpenReview forum: "Retention Depolarization in Recommender System"
_ACM.org/TheWebConf/2024/Conference — TheWebConf24_

### Official Review · Reviewer_SjAq · 2023-11-21

**Novelty:** 5
**Technical Quality:** 6

**Review:**

This paper studies long-term fairness in recommendation systems to mitigate retention depolarization. The authors propose an RL-based framework to ensure long-term retention fairness and recommendation performance.

Strength:
1. Compared with short-term recommendation fairness, long-term fairness is less investigated. This paper focuses on the novel long-term unfairness issue of retention depolarization, which is of practical significance.
2. The proposed method achieves better long-term recommendation performance and fairness simultaneously compared with a wide range of baselines including short-term fairness baselines and RL-based heuristic variances.
3. In addition to empirical results, the authors also conduct theoretical analysis.

Weakness:
1. The paper frequently mentions “groups” and “retention fairness” but these concepts are not clarified until much later. A clearer and more upfront definition of these terms could greatly benefit the reader's understanding, particularly before delving into the methodology section.
2. Some notations and descriptions can be improved. (1) On page 2 when introducing the MDP, what is the initial user state distribution in detail; (2) The notation “a” is used in different places with diverse meanings (e.g., group, sampled item), it would be better to use different notations for clarity.

**Questions:**

Refer to weakness
Other questions:
Q1: Is there any guideline on how to set w in equation (2)? Although ideally, we can adjust w for different tradeoffs, how to choose it in practice.
Q2: Equation 2 uses the difference between group retention as a disparity constraint, in the final evaluation, fairness is defined as the ratio of group retention probability. Is there any particular reason behind this choice?
Q3: In the evaluation, how to ensure that the environment simulator is well-trained such that it mimics the practical setting.

**Reviewer Confidence:**

2: The reviewer is willing to defend the evaluation, but it is likely that the reviewer did not understand parts of the paper

**Scope:**

4: The work is relevant to the Web and to the track, and is of broad interest to the community

---

### Official Review · Reviewer_qDDn · 2023-11-27

**Novelty:** 4
**Technical Quality:** 5

**Review:**

This paper introduces ReFair, a model-based reinforcement learning framework that iteratively improves recommendation policies to maximize cumulative reward while ensuring small retention disparity between user groups over time. It formulates the problem as a constrained MDP and constructs a surrogate optimization problem to optimize policies regarding an estimated environment model and model uncertainty.
Strengths:
- The paper addresses an important problem of longitudinal group fairness in recommendations from the lens of platform polarization. However simple, it seems like the problem is understudied in literature.
- Presents a novel RL formulation and approach (pages 3-4), and contrasts it with other repeated risk minimization (RRM) techniques that are often associated with the domain.
- Provides theoretical guarantees on sublinear regret under linear MDP assumption (pages 11-12).
- Experimental results demonstrate effectiveness on real-world datasets (page 7) and comparisons to known RRM techniques.

Weaknesses:
- Retention depolarization might not always be a positive goal. While optimizing for the two groups to have similar retention rates, the platform could still be highly unfair to individuals of one of the groups. For example, a video streaming platform might engage one group by showing them conspiracy videos while showing the other group healthy content that is also relevant. Or, a dating platform might maximize matches by creating very polarized profiles for some genders that may attract but also antagonize some users, to balance metrics between genders.
In other words, there are alternatives to retain user groups at the cost of exposing them to harmful/unhealthy content, merely to match retention with another group.
- The paper relies on estimating environment models from offline datasets, which may not capture complex real-world dynamics. Previous works with similar contrived setups often show a lack of impact when it comes to applying lessons and techniques to real-world problems.
- The theoretical guarantees and simulator used both assume dynamics can be modeled from finite data, which may not hold in practice.
- The paper only considers fairness between two groups, while multi-group settings were briefly mentioned without details. It seems like the extension will be non-trivial.

**Questions:**

Here are some questions on top of the weaknesses mentioned in the previous response:
- Is retention depolarization always a good goal? See comment above. Also, please comment on the possible drawbacks in case of intersectionality where certain subgroups might be severly impacted by the model's behavior while trying to minimize gap in retention rates. Would adding individual fairness constraints alleviate such issues?
- In the simulator setup, the model is trained on the dataset provided and the events corresponding to user churn. However, this process itself relies on a selection biased dataset where the simulator is not guaranteed to work well on policies that are very different from the policy represented in the datasets. How do you reconcile these differences?
- Did you consider model-free reinforcement learning for this setup? It seems like the experiment setup is more suited for model-based RL but defining a model-free approach would allow for broader applicability.
- Comment in general about the best practice for sim-to-real approach of your method. Did you also consider other ways to learn the simulator, e.g., something that performs a bayesian update of the model whenever new data is obtained.

**Reviewer Confidence:**

3: The reviewer is confident but not certain that the evaluation is correct

**Scope:**

3: The work is somewhat relevant to the Web and to the track, and is of narrow interest to a sub-community

---

### Official Review · Reviewer_kizF · 2023-11-29

**Novelty:** 4
**Technical Quality:** 4

**Review:**

1) The abstract mentions the issue of retention disparities in recommender systems and proposes a framework, ReFair, to address it. However, the problem statement could be more precise regarding the nature of these disparities. It's not clear how these disparities manifest across different domains or types of recommender systems. A more detailed explanation of specific instances or scenarios where these disparities are most evident would provide better context and understanding of the problem's scope.

2) The research assumes that retention fairness is crucial for the long-term success of recommender systems. While this is a reasonable assumption, it would benefit from empirical validation. For instance, demonstrating through case studies or existing literature how retention fairness directly correlates with business metrics or user satisfaction in recommender systems would strengthen the argument.

3) The paper indicates that ReFair involves alternating between environment learning and policy improvement. However, it lacks detailed methodological transparency, especially concerning the algorithms used for these processes. For practical usage and in-depth evaluation, the paper should explicitly detail the algorithm from aspects including any assumptions made, the data structures used, and the computational complexity involved.

4) The paper's methodology assumes a relatively stable set of user preferences over time. This assumption may not hold true in real-world scenarios, where user preferences can change rapidly due to various factors like trending topics, seasonal effects, or personal life changes. This variability in user preferences can significantly impact the effectiveness of the proposed approach. The paper should address how their model adapts to these changing user preferences and whether it can dynamically adjust its recommendations to reflect these shifts.

5) The paper discusses addressing retention fairness between two user groups, but does not delve deeply into the dynamics that might exist within and between these groups. In reality, user populations are often more complex, with intersectionalities based on multiple user attributes like age, location, interests, etc. A more nuanced understanding and approach to handling these intersectionalities in user groups could enhance the model's effectiveness in ensuring retention fairness.

**Questions:**

1) How precisely do retention disparities manifest across different domains or types of recommender systems, and could you provide specific instances or scenarios where these disparities are most evident?

2) Can you empirically validate the assumption that retention fairness is crucial for the long-term success of recommender systems, perhaps through case studies or existing literature demonstrating its correlation with business metrics or user satisfaction?

3) Could the paper provide more methodological transparency, especially regarding the algorithms used for alternating between environment learning and policy improvement, including their assumptions, data structures, and computational complexity?

4) How does the model adapt to rapidly changing user preferences due to factors like trending topics, seasonal effects, or personal life changes, and can it dynamically adjust its recommendations to reflect these shifts?

5) How does the model address the complex dynamics and intersectionalities within and between user groups, based on multiple user attributes like age, location, and interests, to enhance its effectiveness in ensuring retention fairness?

**Reviewer Confidence:**

3: The reviewer is confident but not certain that the evaluation is correct

**Scope:**

4: The work is relevant to the Web and to the track, and is of broad interest to the community

---

### Official Review · Reviewer_98qg · 2023-12-01

**Novelty:** 6
**Technical Quality:** 3

**Review:**

The paper studies the question of long-term performance of recommender system through the lens of user retention. They propose a model-based RL approach to maximizing the long-term average user satisfaction under the constraint that "retention" is kept similar across groups of users at every time step.

The authors propose some theoretical arguments, algorithms and perform experiments on simulated environments to study their algorithm, comparing to relevant baselines.

Overall,  like the idea of retention-fairness, which is an interesting concept, which makes sense both from  perspective of fairness and the perspective of long-term total(sum) user satisfaction.

On the other hand, I have a lot of questions regarding the mathematical treatment and experimental results, and at this stage I find at least the paper confusing in the details. I might review my ratings if there are convincing answers from the authors, but there may also be too many changes for the paper to be accepted without a second round of reviews.

**Questions:**

questions:
- section 3: the model assumes a user leaves the platform forever. How different would it be if we assume that users can "come back" with small probability? I would expect the overall problem to be easier
- section 3: it seems that we cannot have incoming users, which means that we will never have more uses than at the initial time. This seems a more difficult scenario than in real life, and it ignores what we should/could do to get new users in the system.
- RRM naturally allows to deal with non-stationarities (in items or user interests). How would nonstationarities be accounted for? What would be the interplay between non-stationarities and optimizing for long-term performance?
- equation 2: is the problem always feasible? If a group is intrinsically more difficult to retain than another (e.g., because we don't have the right items), then equalising retention doesn't seem possible
- maybe it would be good to have an inuition/motivation example explaining why maximizing retention of every group yields superior policies to maximizing the minimum average reward in each group
- "once a user abandons the platform, there is no way for the system to get the user back. In other words, our problem is not episodic: once initiated, we can never restart from the initial state.Thus, we appeal to a model-based RL solution" -> it's not obvious to me that model-based RL is a solution: since it takes some time to learn the model, we still cannot get back the users who left while we were training the model.
- "Let pi_t represent the optimal policy for Eq. 2" I'm confused here. equation 2 only defines "pi_theta" in some way that doesn't depend on t.  How do you define pi*_t? in particular, pi*_t depends on pi*_{t'} for t' in {t+1,t+T}. In equation 2, how do you define the cumulative sum of rewards as a function of pi_theta?
- "And later we prove that solving this surrogate optimization problem leads to sublinear regret in recommendation performance and sublinear cumulative fairness constraint violation." -> how is it possible to have sublinear regret since any user lost because of exploration (e.g., through the bonuses) can never be recovered?
- how rerasonable/practical are the linearity assumptions? If the state repesentations are learnt, what is the cost of learning them?
- lemma 3.3 (regarding feasibility of the problem with epsilon=0) under what conditions does c_t -> 0? Theorem 3.4 seems to say that it is always the case, but I don't see why it should be possible to always equalize retention without additional assumptions on user groups (e.g., what prevents a group to always stay whatever you do and another group to always leave whatever you do)
- how do we define V^* in some way that doesn't depend on t in the definition of regret?
- since a user who leaves the platform is counted as 0 reward at each time step, it is not clear to me why we need to explicitly account for retention -- intuitively, maximising the long-term reward should be enough to encourage retention since the long-term cost of users who leave is huge.
- ReFAIR is better than RL-unfair and ReFAIR-onlyQ on the cumulative reward of Figure 1 and Figure 2. why is the so? Since both RL-unfair and ReFAIR-onlyQ are suppposed to maximize cumulative reward without the retention equality constraints, they should be better on cumulative reward than refair

**Reviewer Confidence:**

3: The reviewer is confident but not certain that the evaluation is correct

**Scope:**

3: The work is somewhat relevant to the Web and to the track, and is of narrow interest to a sub-community

---

### Official Review · Reviewer_Gzst · 2023-12-01

**Novelty:** 5
**Technical Quality:** 6

**Review:**

This paper studied the polarization problem within the user population in recommendation systems. Starting from the retention disparities issue among user groups, they proposed a novel fairness notion concerning long-term retention across different user groups and designed the framework that enables continuous improvement of recommendation algorithms while enforcing retention fairness.

Strengths:
1. This work is the first to study improving recommendation performance while enforcing retention fairness over time.
2. The paper is well-written with easily understandable notations and a clear presentation of well-motivated method design. The theoretical analysis is solid with detailed proof in the appendix.
3. The proposed retention disparity is very interesting to the community I think.
4. The experiments have shown the effectiveness of the proposed REFAIR compared with other RRM and RL baselines. The ablation study shows the effects of some core parts in the optimization goal.

Weaknesses:
1. There is some little concern about the notations as shown in questions.
2. Could you discuss the gap between your theoretical analysis based on your assumptions and your practical implementation? Does theorem 3.4 still hold in your practical implementation?
3. In the comparison with baselines, why don't you compare with [7], which I think is a most related work to yours?
4. Others see in questions.

**Questions:**

1. Did you provide the code of your framework? There is no content in the link to the code you gave in the paper on page 2. How do you choose $d$ and $v$ for the two datasets?
2. What is $K$ in lines 401 to 404? Is it the group number (but the symbols are different)?
3. Intuitively,  the relaxation factor $c_t$ should decrease as time progresses and more data is collected as you mentioned in lines 323-325. However, from the formulation of valid compatible $c_t$ in Lemma 3.3, can we still make this statement?

**Reviewer Confidence:**

3: The reviewer is confident but not certain that the evaluation is correct

**Scope:**

4: The work is relevant to the Web and to the track, and is of broad interest to the community

---

### Decision · Program_Chairs · 2024-01-22

**Decision:**

Accept

**Comment:**

Our decision is to accept. Please see the AC's review below and improve the work considering that and the reviewers' feedback for cemera-ready submission.

"Reviewers praised this paper for study the novel problem of fair retention over time, and agreed that the work was relevant to TheWebConf. Multiple reviewers were confused about whether the method could handle non-stationary preferences, so the authors should work to clarify these concerns in the final paper. Overall, though, I think the authors have adequate responded to the reviewers' questions and recommend this paper for acceptance."